

# VOC sources and impacts at an urban Mediterranean area (Marseille – France)

Marvin Dufresne[1,2], Thérèse Salameh[1], Thierry Leonardis[1], Grégory Gille[3], Alexandre Armengaud[3], Stéphane Sauvage[1]

[1]: IMT Nord Europe, Institut Mines-Télécom, Univ. Lille, Centre for Energy and Environment, F-59000 Lille, France
    [2]: French Environment and Energy Management Agency 20, avenue du Grésillé - BP 90406 49004 Angers Cedex 01 France
    [3]: Atmosud, Le Noilly Paradis - 146 rue Paradis, 13006 Marseille, France

*Correspondence to*: Marvin Dufresne (marvin.dufresne@imt-nord-europe.fr)

**Abstract.** Long-term measurements of VOC concentrations are crucial to improve our knowledge about their role in

atmospheric chemistry, especially in region with high photochemistry such as the Mediterranean Basin. A field measurement campaign of 18 months has been conducted in Marseille from March 2019 to August 2020 with online measurement of C2 to C16 NMHC using two TD-GC-FID instruments. The positive Matrix Factorization model has been applied to this dataset for each season. Six factors were identified yearlong (traffic exhaust, fuel evaporation, industrial source, shipping, regional and local urban background and IVOC) and two were identified as seasonal factors (biogenic in summer and residential heating

during cold period).

The traffic (exhaust and evaporation) is the first contributor to NMHC concentration measured with a relative contribution of about 40 % with the exception of spring 2020 where the relative contribution was only 25 %. The potential contribution of each factor to secondary pollutants formation has been evaluated. Results reveal that the shipping source is potentially one of the most important contributors to the Secondary Organic Aerosol formation potential despite the low contribution of this

factor to NMHC concentration.

The impact of the lockdown due to Covid-19 is clearly visible on all sources and especially on the traffic source. The contribution of this source has decreased by a half during spring 2020 in comparison with other seasons.

A comparison of these results with emission inventories should be useful to evaluate their accuracy for a better understanding of the atmospheric pollution occurring at Marseille.



## 1 Introduction

Non-Methane HydroCarbons (NMHC) are emitted into the atmosphere by different sources either of anthropogenic origin like road traffic, residential heating, solvent use and industrial activities or of natural origin like biogenic emissions from plants, trees, etc. (Wang et al., 2020; Panopoulou et al., 2018; Thera et al., 2019). Once emitted, these compounds participate
in the tropospheric photochemistry contributing to the formation of secondary pollutants like ozone and Secondary Organic Aerosols (SOA) which may impact health and climate (Lelieveld et al., 2015).

Some of these NMHC like benzene and 1,3-butadiene are classified as carcinogenic by the International Agency for Research on Cancer (IARC). Besides the health impact of some of these NMHC, secondary pollutants have also an impact on health. A recent publication from the World Health Organization (WHO) has pointed out that there are 4.2 million of
people dying of diseases each year linked to ambient air pollution and 9 persons out of 10 in the world breathe polluted air (World Health Organization, 2018).

Air pollution is even more dramatic in urban environment with, according to the WHO more than 80 % of people living in urban areas where air pollution exceeds the WHO air quality guidelines (World Health Organization, 2016). Some urban environments like the Mediterranean basin are particularly affected by ambient air pollution and its consequences due to the
growing urbanization in this region, as well as due to considerable anthropogenic and environmental pressures, with the contribution of important sources from industries and shipping. Additionally, the Mediterranean climate, with high photochemistry, is favorable to the formation of secondary pollutants which makes this region a hot spot of global warming (Lelieveld et al., 2014). The ChArMEx project (Chemistry - Aerosols Mediterranean Experiments) through a coordinated experimental effort, aimed at assessing the budget and the climatology of atmospheric pollutants and their impact on air
quality and climate. The associated project TRANSEMED (TRANSport, Emissions and Mitigation in the East Mediterranean) focused on atmospheric pollution due to anthropogenic activities in urban areas of the Eastern part of the Mediterranean basin. In this frame, studies have shown high level of organic compounds in Beirut, Athens and Istanbul (Panopoulou et al., 2018; Salameh et al., 2016; Thera et al., 2019). Concerning the western part, a study has been performed in Cape Corsica at a remote site (Michoud et al., 2017), but there is a lack of studies on urban areas in this part of the basin.
Marseille, a French city in the western part of the Mediterranean basin is of interest to improve our knowledge concerning air pollution in this urban area of the basin. Due to its proximity with an important industrial complex, a part of the port inside the city and an important anthropogenic activity (the second city and the third metropole in France in term of number of inhabitants) Marseille is particularly affected by air pollution.



In Marseille, frequent ozone pollution episodes are observed in summer while PM pollution events occur mainly in winter (Chazeau et al., 2021). The significant health impact of ambient air pollution has been demonstrated in Marseille (Khaniabadi and Sicard, 2021; Magazzino et al., 2020).

Despite these secondary pollutant pollution events, there are still few studies concerning NMHC in Marseille and the last one occurred 20 years ago and only during few weeks (Coll et al., 2010). According to CITEPA (CITEPA, 2022), some NMHC source emissions have highly decreased in the last years. This decrease may be visible on NMHC concentrations (Waked et al., 2016), but uncertainties associated to NMHC emission inventories are still high especially regarding their speciation (Thunis et al., 2016; Trombetti et al., 2018). Moreover, heavier NMHC like Intermediate Volatility Organic Compounds (IVOC) are not well investigated in the literature despite their important contribution to the SOA formation (Ots et al., 2016) and have not been investigated in Marseille yet. This lack of knowledge increases uncertainties of local and regional emission inventories that are used as inputs in the chemistry-transport models used for the implementation of policies to prevent air pollution episodes.

The aim of this work is to identify and quantify the major sources of NMHC at an urban background site in Marseille-France over a one year and a half dataset and evaluate their impact on air pollution. A large set of speciated NMHC from C2 to C16 including IVOC has been continuously measured on an hourly basis from March 2019 to August 2020 providing a unique database.

## 2 Experimental procedure

### 2.1 Measurement site

A field campaign was conducted from March 2019 to August 2020 at "Marseille – Longchamp", an urban background station operated by the local Air Quality Monitoring Network (AASQA) - AtmoSud. This measurement station is located in a residential area, in the Longchamp Park more precisely, at 43°18'19" latitude north, 5°23'41" longitude east and at an altitude of 71 m above sea level. The measurement site (Fig. 1) is at 2 km of the port in the east-southeast and at 100 m of the nearest road axis. Industrial sources are located farther: a plastic industry at 9 km eastern the site, metallurgical manufactory and industrial port of Fos-Marseille at 40 km in the west-northwest, and a petrochemical industry at 27 km in the northwest. The airport is located at 20 km in the northwest.

Marseille has a typical Mediterranean climate with rainy winter and dry summer. Meteorological conditions during the campaign will be discussed in results and discussion section.





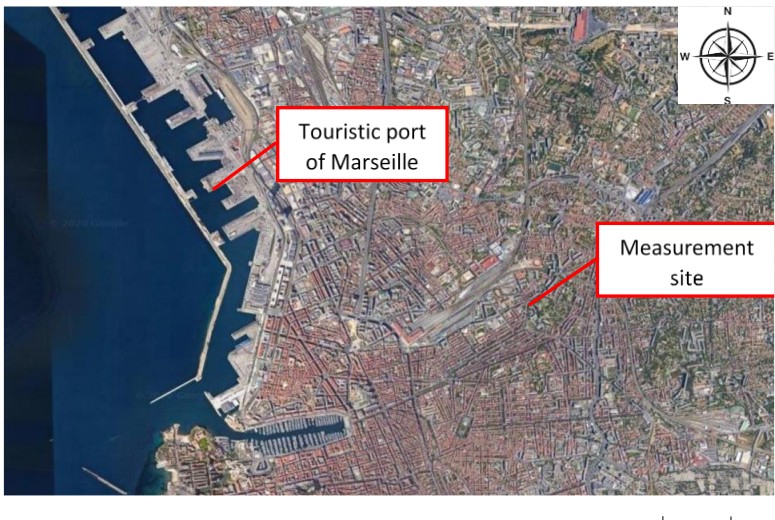

**Figure 1: The measurement site in Marseille. Adapted from © Google, 2020, TerraMetrics.**

## 2.2 Material and methods

Two TD-GC-FID (Thermal Desorber-Gas Chromatography-Flame Ionization Detector), from Perkin Elmer (Perkin-Elmer,
Waltham, USA) and Chromatotec (Val-de-Virvée, France), have been implemented at the measurement site allowing the
hourly measurement of 70 NMHC (Non-Methane HydroCarbons) from 2 to 16 carbon atoms. Perkin-Elmer instrument (TD-
GC-2FID) measures a total of 61 NMHC with a number of carbon atoms between two and ten thanks to the two FID.
Chromatotec instrument allows the measurement of a total of 44 C6 to C16 NMHC. The combination of the measurement of
these two devices gives us the mixing ratio of 35 common compounds.

Compounds are pre-concentrated in a trap and transferred to the chromatographic columns via a heated transfer line.
Technical details of the devices are given in Table 1.

A nafion dryer has been added in the sampling line of the TD-GC-2FID to remove water as it can affect NMHC
measurements by increasing their concentration up to a factor of 2.2 for some compounds like ethene (Bourtsoukidis et al.,
2019). In addition an ozone scrubber made of a 15 cm KI-coated copper tube, prevented ozone effect on the most reactive
species like alkenes (Bourtsoukidis et al., 2019).

A certified standard prepared by the NPL (National Physical Laboratory, Teddington, England) is used to calibrate both
devices during the whole campaign, including 29 NMHCs at 4 ppb level. For compounds that are not present in the NPL
standard, the theoretical response factor of FIDs has been calculated by using a reference compound and the effective carbon
number (ECN) (Dietz, 1967; Sternberg et al., 1962).




**Table 1: Characteristics of installed instruments.**

|  | TD-GC-2FID | TD-GC-FID |
|---|---|---|
| **Manufacturer** | Perkin-Elmer | Chromatotec |
| **Sampling flow** | 25 mL.min⁻¹ | 63 mL.min⁻¹ |
| **Trap composition** | 100 mg of carbosieve SIII and 20 mg of carbopack B | Unknown composition ** |
| **Trap temperature** | Min : -30°C<br>Max : 300°C | Min : ambient<br>Max : 380°C |
| **Column composition** | Column A : CP Sil 5CB (50m × 0,25mm × 1µm)<br>Column B : Plot $Al_2O_3/Na_2SO_4$ (50m × 0,32mm × 5µm) | Metallic capillary column MXT30CE (30m × 0,28mm × 1µm) |
| **Detection limit** | 10 – 16 ppt * | 5 – 12 ppt |

*: Except for ethane, ethene, propane, propene and acetylene which have respectively a detection limit of 104, 45, 102, 35 and 65 ppt.

**: This information is not communicated by the supplier.

## 2.3 IVOC Calibration method

In the absence of a calibration standard for IVOC measured by the C6 – C16 TD-GC-FID, a calibration procedure has been developed to check the relevance in using ECN. A liquid mixture containing all linear alkanes from C6 to C16 and toluene is injected in two evaporation systems: (i) a heated calibration solution rig, normally used for doping sorbent tubes, has been

adapted for the transfer of a gaseous mixture into the TD-GC-FID, (ii) and a Liquid Calibration Unit (LCU) from Ionicon® (Innsbruck, Austria) dedicated to the transfer of gaseous mixture from an initial liquid mixture.

Only hexadecane showed significant differences (30% underestimation in comparison with theoretical results from ECN for the heated calibration solution rig and 15% for the LCU. The other compounds are all below 10% with both evaporation systems) where a special attention on its concentration should be considered. See the supplement part. Results of the IVOC

Calibration procedure for more details.

During the whole campaign, the running time of both instruments was higher than 80 % leading to a unique hourly dataset of C2 – C16 NMHC for one year and half.



Besides these instruments, other analyzers were running for the measurement of $NO_x$, $O_3$, $SO_2$, $CH_4$ and $CO_2$, as well as an aethalometer for the measurement of black carbon concentrations and a particle counter. Meteorological parameters such as

temperature, pressure, wind speed and wind direction complete the dataset.

### 2.4 Quality assurance/Quality control

During the whole campaign many Quality assurance/Quality control (QA/QC) tests have been performed to ensure the quality of the dataset, based on ACTRIS (Aerosols, Clouds and Trace gases Research InfraStructure) guidelines (Reimann et al., 2018). Both instruments repeatability, reproducibility, and blank, have been checked many times during the campaign.

Comparison have also been made between both instruments on common species and for a same instrument between isomers. More details are available on the supplementary material in the quality control section.

### 2.5 Source apportionment by positive matrix factorization (PMF)

### 2.5.1 PMF model description

EPA PMF 5.0 model was applied to our NMHC dataset. This statistical model has been thoroughly described elsewhere

(Paatero, 1997; Paatero and Tapper, 1994). In this study, the dataset has been separated into six seasons (from spring 2019 to summer 2020) analyzed separately with the PMF.

An input dataset is considered by the PMF as a matrix X of n concentration values and m compounds. The objective of the PMF is to solve the chemical mass balance between measured compounds concentrations and source profiles following the Eq. (1) where the matrix $G(n \times p)$ is the factor contribution, $F(p \times m)$ the factor profile, p the number of factors, n the

measurements, m the species and $E(n \times m)$ the residual part.

$$X = F(p \times m).G(n \times p) + E(n \times m) \qquad (1)$$

The solution of this equation is given by minimizing the residual sum of square Q (see Eq. (2) for Q definition) until the value of Q is converging into a similar minimum for F and G.

$$Q(E) = \sum_{i=1}^{n} \sum_{j=1}^{m} \left(\frac{e_{ij}}{s_{ij}}\right)^2 \qquad (2)$$

$s_{ij}$ is the uncertainty on the measurement for the species i in the sample j and $e_{ij}$ is the residual. Depending on their behavior each compound will be separated into the p factors.

### 2.5.2 Uncertainties calculation

As shown on the equation above, the uncertainty is a prerequisite input for the PMF analysis. Hence, for each measurement from both devices, uncertainties have been calculated using the methodology proposed by the pan-European research





infrastructure ACTRIS. The calculation is based on the sum of two types of errors, random error due to the precision of the device and systematic error. The details of the uncertainties calculation is explained by Hoerger et al., 2015.

Only compounds that are in the standard gas mixture have their uncertainties calculated following this methodology. It represents 28 compounds for the TD-GC-2FID and 13 compounds for the TD-GC-FID. The mean expanded relative uncertainty of these compounds is given in the supplement material Table S3 for both devices. For the TD-GC-2FID the

mean uncertainty range is between 15 and 40 % for the majority of NMHC, and 4 of the compounds have a mean expanded relative uncertainty exceeding 100 %. This is due to concentrations close to the Limit of Detection (LoD) most of the time during the campaign. For the TD-GC-FID, the mean expanded relative uncertainty range is between 5 and 25 % for the majority of NMHC. For the other compounds their absolute uncertainties are estimated by applying the median relative uncertainty of a compound with similar concentration and variability present in the standard gas mixture.

**2.5.3 Dataset missing values and limit of detection**

During the campaign some data were missing or below the LoD. These data must be replaced as missing values are not accepted by the PMF. Concerning concentrations below the LoD, missing values are replaced by the LoD divided by 2 and uncertainties are calculated following the Eq. (3):

$$U = \frac{5}{6} \times LoD \qquad (3)$$

Where U is the uncertainty (in µg.m$^{-3}$).

The missing values are replaced with the hourly median of the month where there are missing values. In this case, the uncertainty associated is the hourly median previously calculated multiplied by 4.

**2.5.4 Determination of the optimal solution**

For each season several base runs were performed with a number of factors varying from 3 to 12. Several parameters have

been plotted versus the number of factors to determine the best solution following the method from Lee et al., 1999 and Hopke, 2000.

An additional parameter F Peak has been used to optimize solutions while checking the rotational variability.

As results, six factors were selected from spring 2019 and 2020 and summer 2020 while seven for the other seasons. All quality indicators are summarized in Table 2. The "bootstrap" test results showed determination coefficients above 0.6 for all

seasons reinforcing the robustness and stability of PMF results.



**Table 2: Mathematical diagnostic for the PMF results.**

| | Spring 2019 | Summer 2019 | Fall 2019 | Winter 2020 | Spring 2020 | Summer 2020 |
|---|---|---|---|---|---|---|
| **n (sample)** | 1034 | 1528 | 2183 | 1706 | 2208 | 1201 |
| **m (species)** | 62 | 59 | 57 | 56 | 42 | 54 |
| **k (factors)** | 6 | 7 | 7 | 7 | 6 | 6 |
| **Q (model)** | 57532 | 79043 | 108751 | 83202 | 102290 | 56069 |
| **$NMHC_{modeled}$ vs $NMHC_{measured}$ ($r^2$)** | 0.874 | 0.924 | 0.905 | 0.896 | 0.909 | 0.937 |
| **F Peak** | -0.5 | -1.5 | -2.5 | -4 | -2.5 | -2 |
| **Mean ratio (modeled vs measured)** | 0.93 | 0.96 | 0.92 | 0.92 | 0.90 | 0.84 |
| **Number of species with $r^2 > 0.75$ for modeled vs measured** | 15 | 19 | 15 | 20 | 16 | 7 |
| **"Bootstrap" minimum (in %)** | 80 | 99 | 94 | 96 | 93 | 79 |

**3 Results and discussion**

**3.1 Meteorological conditions**

The meteorological conditions during the campaign are given in Table 3 on a seasonal basis. During the campaign, three wind directions were dominant. One originates from the north-northwest corresponding to the Mistral, a typical wind of the region characterized by a dry and high wind speed. The two other dominant winds are from the east/east-northeast and the

west/west-southwest and are corresponding to the land – sea breeze circulation. The land breeze is occurring during nighttime with lower temperature than during daytime and then a lower Planetary Boundary Layer (PBL). Furthermore, in land breeze conditions, air masses are coming from land with potential NMHC emissions whereas in sea breeze conditions, air masses are coming from the sea and are poorly polluted. The Figure 2 shows the wind rose during all the campaign.

**Table 3: Meteorological conditions of the measurement site between the 21/03/2019 and the 31/08/2020.**

| | Spring 2019 | Summer 2019 | Fall 2019 | Winter 2020 | Spring 2020 | Summer 2020 |
|---|---|---|---|---|---|---|
| **Temperature range (°C)** | 6.9 – 24.3 | 13.5 – 34.6 | 5.5 – 28.0 | 5.4 – 19.8 | 3.8 – 25.6 | 14.9 – 31.8 |
| **Mean temperature (°C)** | 15.6 | 24.3 | 17.2 | 11.6 | 15.5 | 23.4 |
| **Max wind speed (m.s$^{-1}$)** | 3.5 | 2.5 | 3.5 | 4.5 | 5.8 | 3.3 |
| **Mean wind speed (m.s$^{-1}$)** | 1.0 | 0.7 | 0.8 | 1.1 | 0.9 | 0.7 |



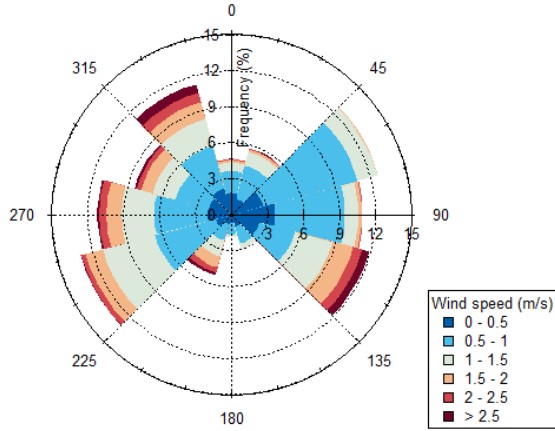


**Figure 2: Wind rose of the entire period of the campaign.**

## 3.2 Impact of photochemistry

The PMF tool is applied to measurements at a receptor site, which can be away from the emission sources. Nevertheless, since the reactivity of some VOC can have an impact on the identification of the sources and their contribution, the
photochemical impact on VOC should be assessed. Recently, a new approach consisting on applying the PMF on reevaluated concentration by estimating the concentration before the photochemistry depletion or the dispersion of the compounds has been applied at many receptor sites (Gu et al., 2023; Liu et al., 2023; Stanimirova et al., 2023; Wu et al., 2023). Here we evaluated the impact of the photochemistry on our dataset by assessing the correlation between reactive species with a less-reactive one (benzene), originating from the same sources, during daytime and nighttime (Salameh et al.,
2015 or Sommariva et al., 2011) in winter and summer. Ethene and m,p-xylene were selected as reactive species whereas n-pentane was chosen as a less-reactive species. The $k_{OH}$ of these compounds are $8.51\times10^{-12}$ molec.cm$^{-3}$.s$^{-1}$, $2.45\times10^{-11}$ molec.cm$^{-3}$.s$^{-1}$, $1.52\times10^{-11}$ molec.cm$^{-3}$.s$^{-1}$, $1.28\times10^{-12}$ molec.cm$^{-3}$.s$^{-1}$ and $3.80\times10^{-12}$ molec.cm$^{-3}$.s$^{-1}$ for ethene, m-xylene, p-xylene, benzene, and pentane respectively. These values are coming from (Atkinson, 1986) except for n-pentane which value is coming from (Atkinson and Arey, 2003). In our case, m and p-xylenes are not separated so the value of $k_{OH}$ for the m,p-
xylene is the mean of both $k_{OH}$ ($1.99\times10^{-11}$ molec.cm$^{-3}$.s$^{-1}$) of m-xylene and p-xylene.

Figure 3 shows the scatterplot of ethene, m,p-xylene and n-pentane with benzene. Daytime data are concentrations measured between 08:00 and 16:00 UTC whereas nighttime data are concentrations measured between 21:00 and 05:00 UTC.

For all these compounds there is no difference between daytime and nighttime scatterplots with benzene either in winter or in summer. As explained in part 3.1, during daytime the measurement site is mainly under sea breeze conditions, the sea is





located at 2 km. Then we assume that NMHC from anthropogenic activities are freshly emitted when we are under sea breeze conditions. The effect of photochemistry on our dataset is negligible and thus also on our PMF results.

**Figure 3: Scatter plot of (a) ethene, (b) m,p-xylene and (c) n-pentane vs. benzene (in ppb) in winter 2020 (left) and summer 2019 (right) during daytime (red) and nighttime (blue).**





### 3.3 General overview

The Figure 4 shows the monthly variability of NMHC families: alkanes C2-C3, alkanes C4-C5, alkanes C6-C9, IVOC (alkanes C10 – C16), alkenes C2-C3, alkenes C4-C6, aromatics and acetylene in ppb. Alkenes concentrations are invalidated between March 2020 and June 2020 due to technical issues. Light alkanes (C2 – C3 and C4 – C5) are the major compounds.

Their measured concentrations are explaining more than 50 % of the concentration of the sum of NMHC each month of the whole campaign. Highest concentrations are observed during cold period between November 2019 and January 2020 and could be explained in one hand by the importance of NMHC emissions from residential heating in the surroundings, low PBL height, and, in another hand, by the photochemical decay undergone in warm period by such reactive species.

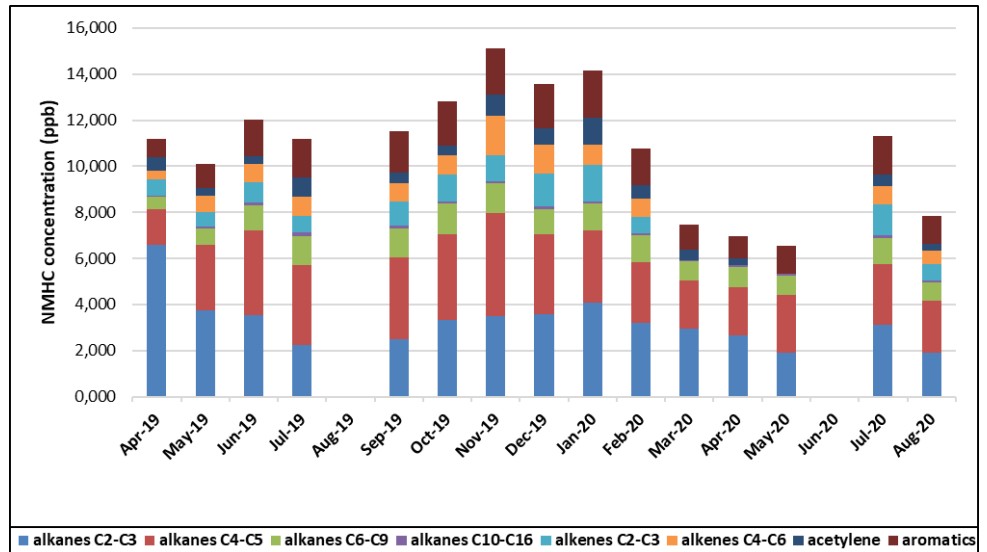

**Figure 4: Monthly variability of NMHC families measured during the one year and half campaign.**

Table 4 shows the comparison of some NMHC concentrations from other studies with the ones obtained in this study in winter and summer. Compared to other Mediterranean cities like Athens or Beirut, NMHC concentrations are drastically lower in Marseille pointing out different country regulations for air quality. On the other hand, measured concentrations at Marseille are lower than those measured in other French cities (Paris, Lyon, Strasbourg) especially ethane, ethylene and

benzene. Additionally, there are some differences between Paris, Lyon, and Strasbourg that might be due to local particularities. Between 2010 (the year of the Paris dataset) and 2019 (the year of the Marseille's dataset) there is a decrease of 20 % of NMHC emissions according to the CITEPA (CITEPA, 2020) and this trend is more or less visible depending on NMHC species (Waked et al., 2016). The most affected sector by this decreasing trend is road transport which is an emitter of alkenes, acetylene, alkanes from C4, and aromatics (Baudic et al., 2016).



**Table 4: NMHC concentration (in ppb) comparison between French cities and cities in the Mediterranean Basin.**

|  | Beirut winter 2012 (Salameh et al., 2015) | Beirut summer 2011 (Salameh et al., 2015) | Athens winter 2016 (Panopoulou et al., 2018) | Marseille winter 2020 (this study) | Marseille summer 2019 (this study) | Paris 2010 yearlong (Baudic et al., 2016) | Strasbourg Yearlong (2003 – 2013) (Waked et al., 2016) | Lyon yearlong (2007 – 2013) (Waked et al., 2016) |
|---|---|---|---|---|---|---|---|---|
| Ethane | 2.8 | 1.6 | 4.5 | 2.4 | 2.1 | 3.8 | 3.0 | 4.1 |
| Ethylene | 2.1 | 3.3 | 4.1 | 1.1 | 0.6 | 1.6 | 2.1 | 3.3 |
| Acetylene | 2.2 | 2.3 | 4.2 | 0.7 | 0.6 | 0.5 | 0.8 | 0.8 |
| Isopentane | 7.0 | 8.3 | 4.7 | 0.6 | 1.1 | 1.8 | 0.7 | 1.1 |
| Benzene | 0.5 | 0.6 | 0.8 | 0.6 | 0.1 | 1.1 | 1.0 | 1.5 |
| Toluene | 2.2 | 3.8 | 2.2 | 0.5 | 0.5 | 0.8 | 0.3 | 1.4 |
| Ethylbenzene | 0.3 | 0.5 | 0.4 | 0.1 | 0.1 | / | / | / |
| m,p-xylenes | 0.9 | 1.8 | 1.2 | 0.3 | 0.4 | / | / | / |
| o-xylene | 0.3 | 0.6 | 0.4 | 0.1 | 0.1 | / | / | / |

### 3.4 PMF factor identification and contribution

Similarly, for all seasons five PMF-factors have been identified related to the sources traffic exhaust, fuel evaporation, industrial, shipping, local and regional urban background and one factor associated to IVOC. In addition, two season-specific factors have also been identified as related to biogenic source in summer and to residential heating during cold period. Details of each source are reported below.

### 3.4.1 Traffic exhaust

In Marseille the car fleet is characterized by an important part of vehicle running on diesel (71.5 %) and a part of 10 % of two-wheelers according to AtmoSud. The proportion of two-wheelers is similar to the one in Paris comprised around 10 and 20 % (Baudic et al., 2016; Salameh et al., 2019).

The factor defined as traffic exhaust is characterized by an important part of aromatic compounds, C7-C8 alkanes and to a lesser extent by C2-C3 alkenes. For each season, around 50 % of the variability of toluene and 1,2,4-trimethylbenzene and 40 % of the variability of isooctane and heptane are explained by this factor. The variability of xylenes and C2-C3 alkenes is explained by this factor by 30 % and 15 % respectively (Fig. 5). Same compounds have been found in traffic exhaust profiles from other studies (Panopoulou et al., 2021; Salameh et al., 2016). In Panopoulou et al., 2021, aromatics compounds and C6 – C9 alkanes are mainly explained by a vehicular exhaust factor but alkenes are separated into a fuel combustion factor due





to many combustion processes like residential heating and traffic. In this study, alkenes are present in the traffic exhaust factor but they are explained by the residential heating factor during cold period by 50 % (see part 3.4.3). Despite the separation of these two combustion processes, there is a correlation between the traffic exhaust and the residential heating factors with a correlation coefficient R of 0.50 in fall 2019, 0.61 in winter 2020 and 0.31 in spring 2020 when the traffic was 250 low.

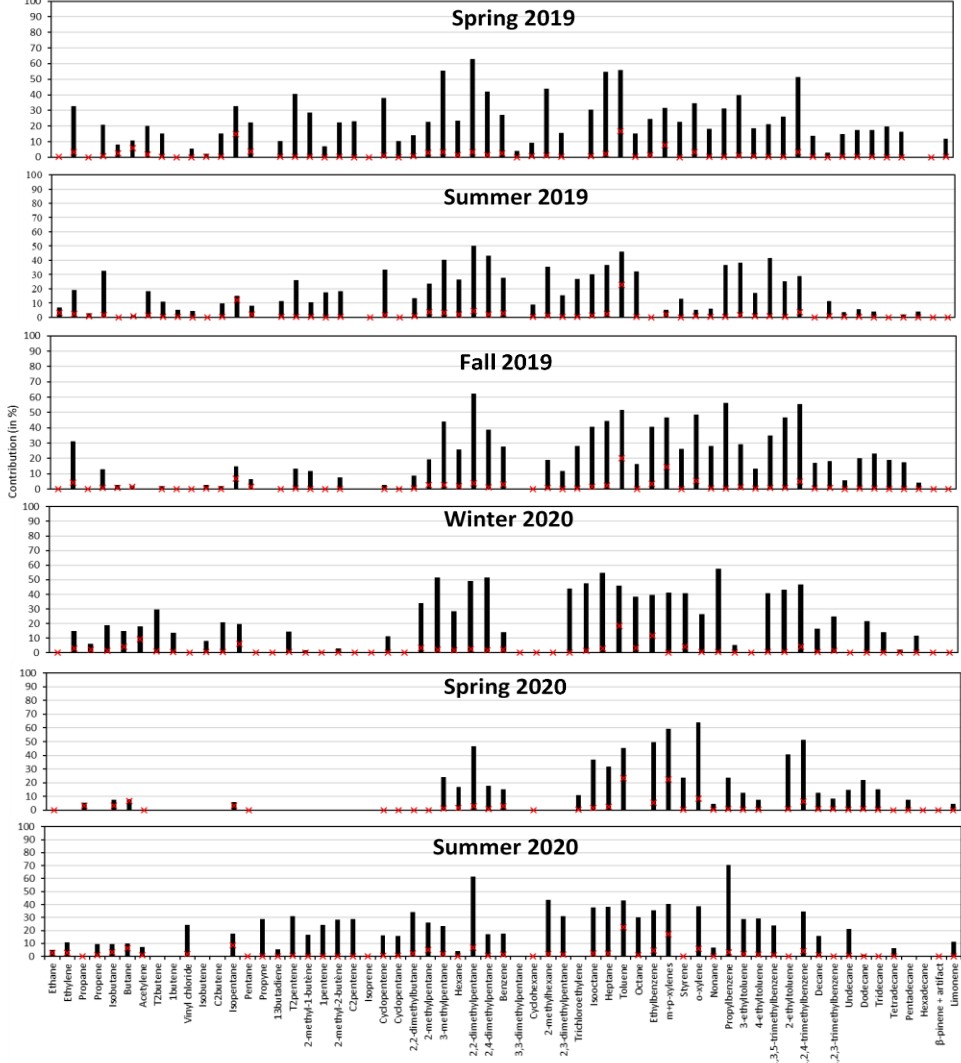

**Figure 5: Traffic factor profile during the whole campaign. The black bars are the contribution of the factor to the species concentration and red crosses are the contribution of the species to the factor.**




More generally, C2 – C3 alkenes are good tracers of combustion processes (Parrish et al., 2009; Warneke et al., 2007) but to distinguish between specific combustion processes other ratios can be verified. For instance, the benzene to toluene (B/T) ratio is a commonly used ratio to determine the origin of the combustion process (Wang et al., 2014 and references therein). A B/T ratio below 0.5 is associated to traffic exhaust and a ratio above 2 to wood or coal burning (Wang et al., 2014). Nevertheless, these values can vary depending on many parameters (type of vehicle, type of wood, regulations, etc…). In the

case of Europe, the benzene is expected to have a lower contribution to the traffic exhaust due to the Directive 98/70/EC for reduction of benzene emission from fuel and solvents (https://eur-lex.europa.eu/eli/dir/1998/70/oj). In this study, the B/T ratio from the factor called traffic exhaust is varying from 0.15 in spring 2019 to 0.07 in summer 2020 pointing out the traffic exhaust origin of this factor on one hand and the low contribution of benzene due to the European directive in another hand.

As shown in Fig. 6, it is worth noting that the traffic exhaust chemical fingerprints obtained from independent PMF analysis are very close regardless the season pointing out the representativeness of this factor in PMF results and supporting the choice to separate PMF analysis by seasons. The PD-SID, which is a statistical test combining Pearson Distance (PD) and Standardized Identity Distance (SID) (Belis et al., 2015; Pernigotti et al., 2016), is largely used in recent PMF studies to evaluate PMF factors (Borlaza et al., 2021; Manousakas et al., 2021; Mooibroek et al., 2022). Here this statistical test has

been applied to the traffic exhaust factor for each season to evaluate a potential seasonal variability of its chemical fingerprint. Results show quite similar fingerprint over seasons (Fig. S4) confirming the conclusion made from Fig. 6.

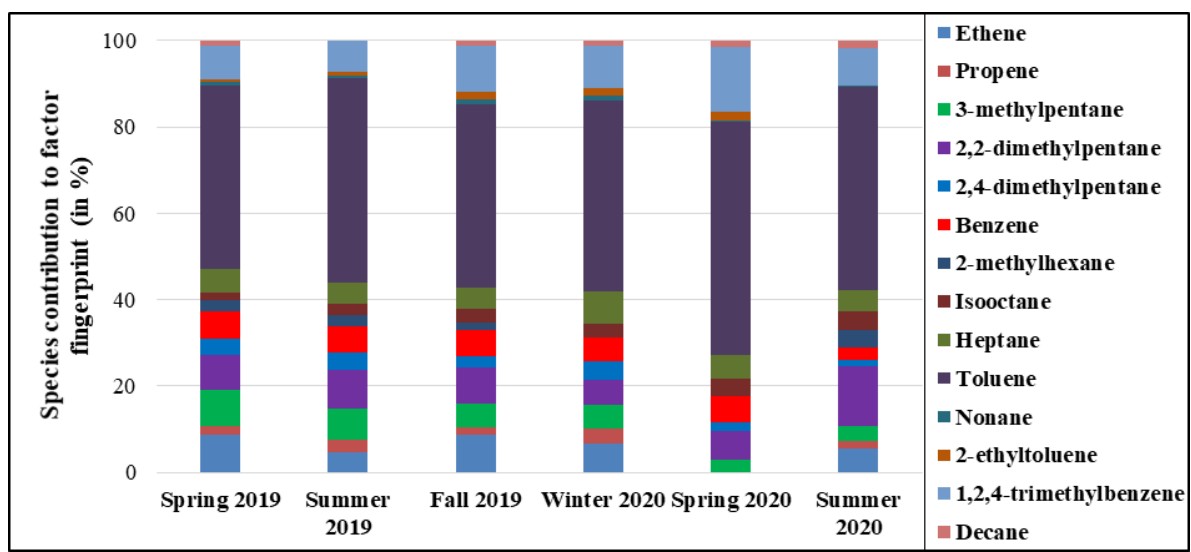

**Figure 6: traffic exhaust fingerprint of 14 NMHC for each season (for technical reasons, ethene and propene were invalidated in spring 2020).**




As tracers of combustion processes, the variability of NO and $NO_2$ are compared with the contribution of this PMF factor. The correlations and the associated p-values (Table 5) are season dependent with R varying from 0.48 in summer to 0.63 in winter for $NO_2$ and from 0.06 in spring to 0.60 in winter for NO. This result may be due to the effect of photochemistry in summer favoring the conversion of NO to $NO_2$ in addition to the regional $NO_2$. The NO is more reactive than the measured

NMHC which probably explain why we don't see any photochemistry impact on our dataset unlike NO. The p-value is always below 1 % showing a significant correlation between $NO_2$ and the traffic exhaust factor but the correlation is not clearly visible for NO due to the distance between the station and the road and the short lifetime of NO. The exception of fall 2019 and winter 2020 can be explained by a lower photochemistry during these seasons.

**Table 5: Pearson correlation coefficient and p-value for the traffic exhaust factor with NO and NO₂ for all seasons. \*\*\* means a p-**
**value < 0.001 %, \*\* a p-value between 0.001 % and 1 % and \* a p-value >1 %.**

| | | Spring 2019 | Summer 2019 | Fall 2019 | Winter 2020 | Spring 2020 | Summer 2020 |
|---|---|---|---|---|---|---|---|
| *NO₂* | **Pearson correlation coefficient** | 0.60 | 0.48 | 0.54 | 0.63 | 0.64 | 0.49 |
| | **p-value (%)** | \*\*\* | \*\*\* | \*\*\* | \*\*\* | \*\*\* | \*\*\* |
| *NO* | **Pearson correlation coefficient** | 0.06 | 0.09 | 0.34 | 0.60 | 0.07 | 0.11 |
| | **p-value (%)** | \* | \*\* | \*\*\* | \*\*\* | \*\* | \*\* |


The average diurnal profile of the contribution of this factor is shown in Fig. 13. One peak is visible at around 07:00 am UTC and a smaller one around 06:00 pm UTC. These peaks are characteristics of traffic emissions corresponding to rush hours and are similar to the ones in other French cities like Dunkirk (Badol et al., 2008) and Paris (Languille et al., 2020). Concerning the seasonal variability, highest values are observed in winter (around 6 µg.m$^{-3}$) and lowest values in summer (around 4 µg.m$^{-3}$) (see Table 11 in part 3.5). This difference could be explained by the i) increase of emissions due to the

cold start of vehicles (Clairotte et al., 2013; George et al., 2015; Ludykar et al., 1999) and ii) the potential photochemical decay of reactive compounds in summer (Filella and Peñuelas, 2006), with an important dilution due to the height of PBL. The same seasonal variability has been observed in Athens, a Mediterranean city, for the traffic exhaust factor (Panopoulou et al., 2021).



### 3.3.2 Fuel evaporation

In urban areas, fuel evaporation originates mainly from two types of sources either from vehicles, either from fuel stations. In the first case, the evaporation source intensity follows the diurnal profile of traffic while it is more scattered in the second case.

In this study, the chemical profile of this source is mainly composed of C4 – C5 alkenes and C4 – C6 alkanes (Fig. 7), explained by 40 to 80 % and 25 to 50 % respectively by this factor. Isopentane represents a key compound of this factor.

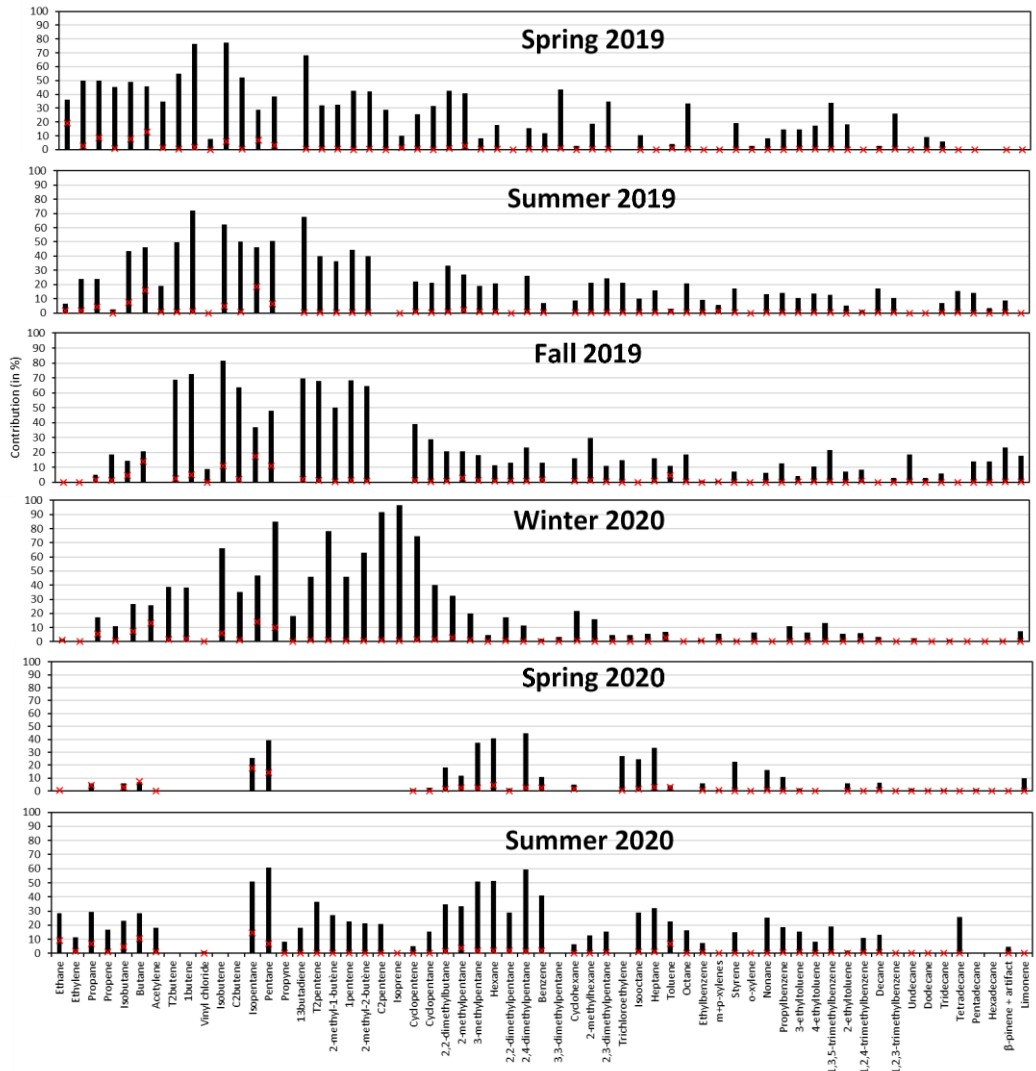


**Figure 7: Evaporation factor profile during the whole campaign. The black bars are the contribution of the factor to the species concentration and red crosses are the contribution of the species to the factor.**



Based on Baudic et al., 2016, high contribution of isobutane and butane was found in fuel evaporation source and C5 – C6 alkanes in traffic exhaust. Other studies, Salameh et al., 2016 and Panopoulou et al., 2021, showed respectively, an

important contribution of C4 – C5 alkanes related to traffic and fuel evaporation, and C5 alkanes in fuel evaporation. In this study, there is a high proportion of C5 – C6 alkanes during all seasons, and a high proportion of isobutane and butane in spring and summer 2019 which is in accordance with fuel evaporation profiles from near field measurements (Salameh et al., 2019).

A fuel station is located at 500 meters in the east of the measurement station and is a potential source of the observed

isopentane concentrations especially when the winds come from the east during land breeze. However, the pollution rose in summer 2019 (in supplement material, Fig. S5) does not show a clear influence of the east wind to this factor's contribution. The seasonal variability of this factor is linked to the temperature with highest values in summer when the temperature is high (around 8 µg.m$^{-3}$) and lowest values in winter (around 6 µg.m$^{-3}$). This seasonal variability is similar to the one of the fuel evaporation related to traffic factor in Panopoulou et al., 2021.

This factor has a correlation coefficient R varying between 0.07 and 0.43 with NO (with the exception of spring 2020 where the value of R is 0.01), and varying between 0.39 and 0.61 with NO$_2$ with a p-value always below 5 % (with the exception of spring 2020 for NO) meaning a significant correlation between the fuel evaporation factor and NO$_2$ and NO to a lesser extent (Table 6).

**Table 6: Pearson correlation coefficient and p-value for the fuel evaporation factor with NO and NO$_2$ for all seasons. \*\*\* means a**
**p-value < 0.001 %, \*\* a p-value between 0.001 % and 1 % and \* a p-value >1 %.**

|  |  | Spring 2019 | Summer 2019 | Fall 2019 | Winter 2020 | Spring 2020 | Summer 2020 |
|---|---|---|---|---|---|---|---|
| *NO$_2$* | **Pearson correlation coefficient** | 0.61 | 0.57 | 0.48 | 0.41 | 0.39 | 0.46 |
|  | **p-value (%)** | \*\*\* | \*\*\* | \*\*\* | \*\*\* | \*\*\* | \*\*\* |
| *NO* | **Pearson correlation coefficient** | 0.12 | 0.12 | 0.43 | 0.32 | 0.01 | 0.07 |
|  | **p-value (%)** | \*\* | \*\*\* | \*\*\* | \*\*\* | \* | \* |

The diurnal profile of this factor (Fig. 13) is similar to the diurnal profile of the traffic exhaust factor. All these findings confirm that this factor corresponds to the fuel evaporation related to traffic. In addition, the contribution of this factor correlates quite significantly (0.41 to 0.61) with the one of traffic-related factor.





### 3.3.3 Residential heating


The factor identified as residential heating is characterized by a large number of light alkanes and alkenes (C2 – C3), acetylene and some aromatic compounds like benzene and ethyltoluenes (Fig. 8). This factor is observed only in fall 2019, winter 2020 and spring 2020. For each of these seasons the variability of light alkanes, alkenes and benzene is explained by more than 40 % by this factor (Fig. 8). The diurnal variation of this factor shows a peak in the morning at 07-08 am UTC and

a peak at 07-08 pm UTC which stays later in the evening (Fig. 13). This factor contribution is independent of a specific wind direction and is explained by the residential area surrounding the station.

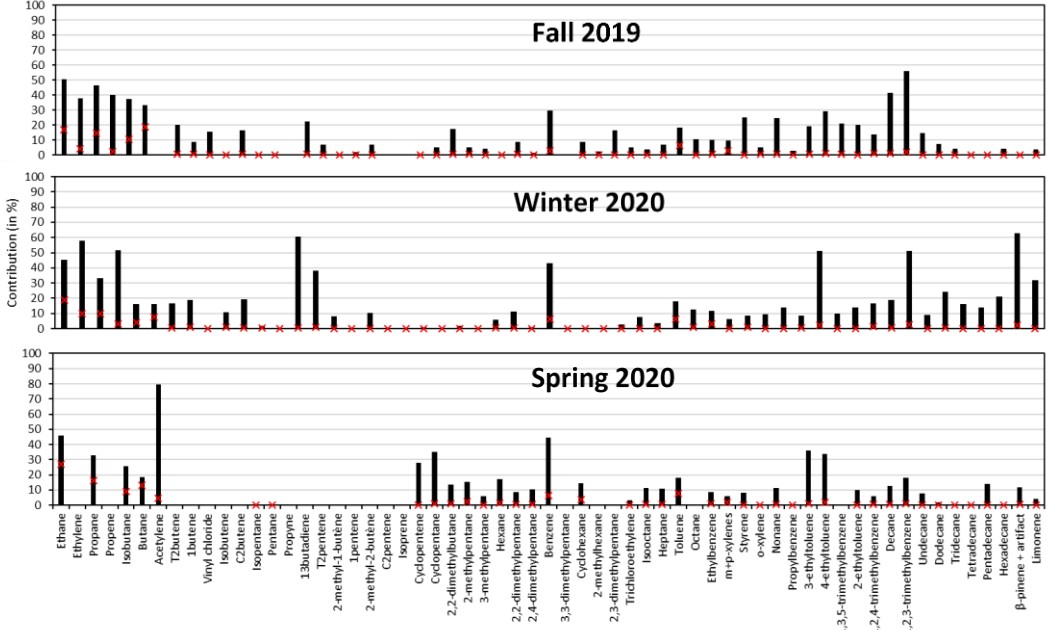

**Figure 8: Heating factor profile during the whole campaign. The black bars are the contribution of the factor to the species concentration and red crosses are the contribution of the species to the factor.**

This profile is similar to other profiles in the literature identified as residential heating (Baudic et al., 2016; Panopoulou et al., 2021; Sauvage et al., 2009). The diurnal profile is barely similar to the wood burning diurnal profile in Baudic et al., 2016 as well as the chemical fingerprint mainly composed of light alkanes and alkenes (C2 – C3), acetylene and benzene confirming the residential heating origin of this factor.

As for traffic exhaust and fuel evaporation, there is a significant correlation with NO and NO$_2$ (around 0.45 and 0.50 for

NO$_2$, and 0.13 and 0.50 for NO). But there is also a high and significant correlation with black carbon from wood burning (0.61 in fall 2019, and 0.80 and 0.78 in winter and spring 2020 respectively) (Table 7).



**Table 7: Pearson correlation coefficient and p-value for the heating factor with NO, NO₂ and black carbon from wood burning for all seasons. *** means a p-value < 0.001 %, ** a p-value between 0.001 % and 1 % and * a p-value >1 %.**

|  |  | **Fall 2019** | **Winter 2020** | **Spring 2020** |
|---|---|---|---|---|
| *NO₂* | **Pearson correlation coefficient** | 0.48 | 0.50 | 0.45 |
|  | **p-value (%)** | *** | *** | *** |
| *NO* | **Pearson correlation coefficient** | 0.50 | 0.42 | 0.13 |
|  | **p-value (%)** | *** | *** | *** |
| *BC_wb* | **Pearson correlation coefficient** | 0.61 | 0.80 | 0.78 |
|  | **p-value (%)** | *** | *** | *** |

According to AtmoSud, in the Aix-Marseille metropole, 18 % of residences are equipped with fireplaces and 6 % of
residences use wood burning as a principal heating source. All other residences are heated with natural gas or electricity. A
low impact of wood burning can therefore occur. To determine an influence of the wood burning emissions to the residential
heating factor, the B/T ratio is used and compared to the one obtained with traffic emissions for the same seasons (Table 8).
For the residential heating factor the B/T ratio is close to 1 in winter and spring 2020 whereas the same ratio is at 0.1 at the
same period for the traffic exhaust factor showing the impact of wood burning. The presence of alkanes C2 – C3 shows that
the residential heating factor is not exclusively explained by wood burning but also by natural gas used as well for residential
heating. Indeed, alkanes C2 – C3 represent important markers of the natural gas used for heating and cooking in cities
(Baudic et al., 2016).

**Table 8: Benzene to Toluene ratio for the traffic factor and the residential heating factor from fall 2019 to spring 2020.**

|  | **Fall 2019** | **Winter 2020** | **Spring 2020** |
|---|---|---|---|
| **Residential heating** | 0.44 | 0.94 | 0.84 |
| **Traffic** | 0.14 | 0.12 | 0.11 |
| R_{heating/BCwb} | 0.61 | 0.80 | 0.78 |

It is worth noting that terpenes (limonene and β-pinene) measured concentrations are mainly explained by the residential
heating factor in winter. It is known that in urban areas terpenes can be emitted by anthropogenic activities (Borbon et al.,
2023; Dominutti et al., 2019; Kaser et al., 2021; Panopoulou et al., 2020). A comparison between limonene and benzene and
toluene in winter shows a clear correlation specially during nighttime. Anthropogenic emission of terpenes from residential
heating cannot be discarded.





### 3.3.4 Industrial

The following factor is characterized by the presence of cyclohexane explained by more than 70 % by this factor, and benzene and isooctane which are explained by around 15 to 35 % by this factor, regardless the season (Fig. 9).

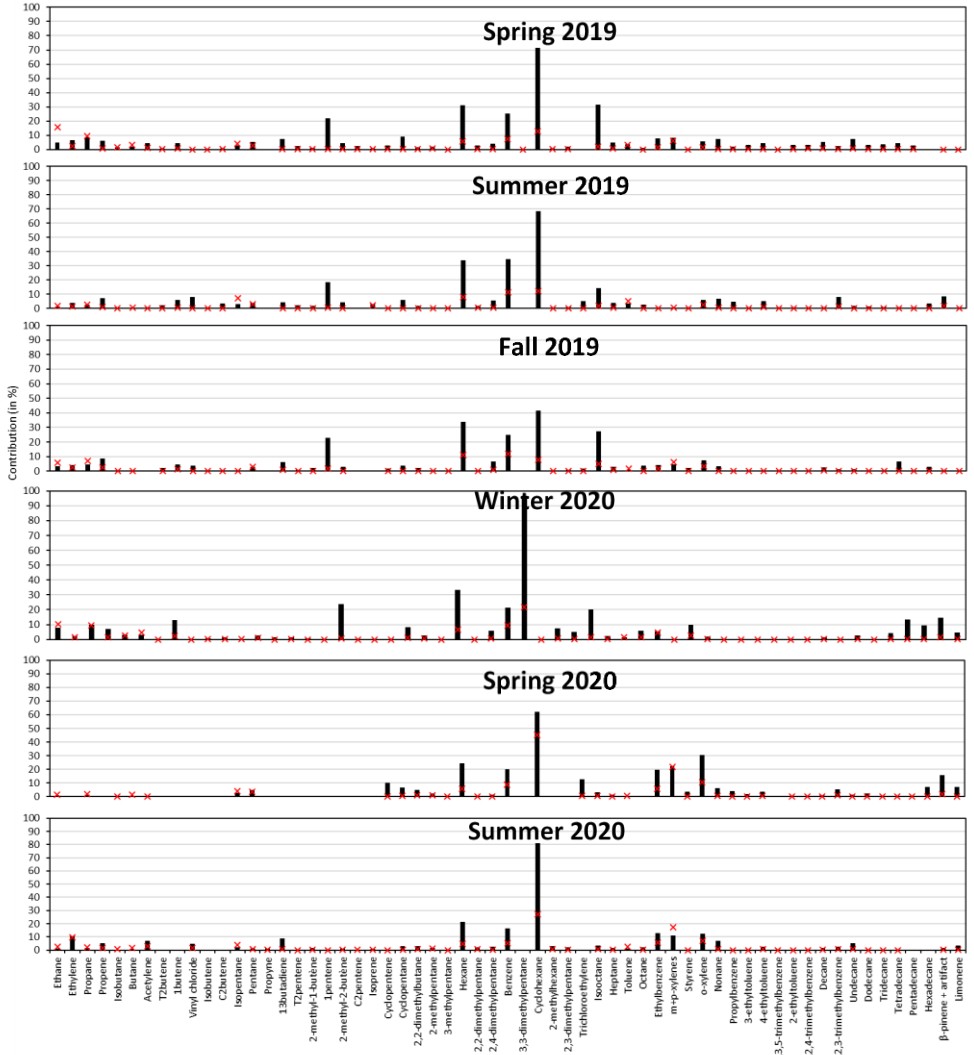

**Figure 9: Industrial factor profile during the whole campaign. The black bars are the contribution of the factor to the species concentration and red crosses are the contribution of the species to the factor.**





The temporal variability is characterized by a low background level and episodic high intensity peaks. Those peaks occurred exclusively during land breeze events corresponding to east wind direction (Fig. 10). This sector points out the presence of a plastic production industry located in the east of the station. This factor is thus attributed to an industrial source.

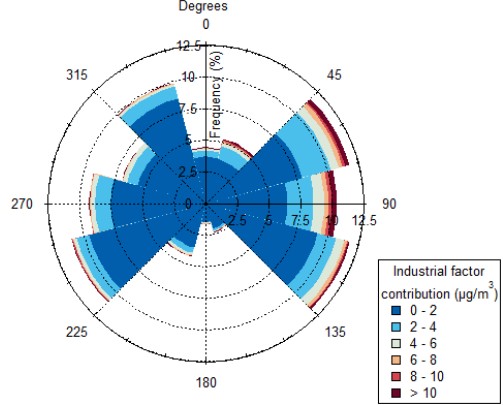

**Figure 10: Pollution rose of the industrial factor during the whole campaign.**

A similar profile with high benzene and cyclohexane concentrations during nighttime has been found during a study held at the station of Marseille Prado (43°27'N latitude, 5°13'E longitude) where the impact of the plastic production industry on NMHC concentrations in Marseille has been pointed out (Troussier et al., 2005).

### 3.3.5 Shipping

A specific factor has been identified during all the seasons in 2019 explaining from 35 to 60 % of the concentration of
xylenes (including o, m and p-xylenes) and ethylbenzene (Fig. S6 in supplement). The contribution's temporal variability of this factor shows low background levels with high peaks regardless the season and regardless the wind direction (Fig. S7 in supplement). This factor shows some correlation with $NO_2$ and traffic exhaust factor (R Pearson above 0.40 with the exception of fall 2019 for both) but has no correlation with NO (table S4). Hence, like traffic related sources and residential heating, this factor seems to be linked to combustion processes. The highest correlation is observed with the IVOC factor in
Spring 2019 with a Pearson correlation coefficient of 0.60 highlighting a possible link between both factors, at least for this season. However, the correlation with the IVOC factor shows strong variation with a maximum of 0.60 in Spring 2019 and a minimum of -0.02 in Fall 2019 (table S4). This variation may be linked to the low concentrations of IVOC measured during the whole campaign.

Furthermore, in the framework of a recent ADEME/AQACIA project called PAREA (Evolution des PArticules fines en
champs proche du tRafic maritimE à Marseille) a measurement campaign took place at the same measurement site of Marseille Longchamp and at the port of Marseille, simultaneously, during summer 2021 for the establishment of the





chemical fingerprint of shipping emissions. This source fingerprint determined within PAREA project, is comparable to the profile of this PMF factor, with the presence of EX (ethylbenzene and xylenes) and, to a lesser extent, toluene (PAREA project report, should be published by the end of the year). The variability of this source characterized by few peaks occurring at random times can be explained by several reasons:

- the VOC emissions of the maritime traffic correspond to only 2 % of the total VOC emissions in Marseille in 2019 according to the AtmoSud emission inventory (https://cigale.atmosud.org/visualisation.php) which is similar to the contribution of this factor to NMHC concentration determined here (between 4 and 8 %),
- only 10 to 20 % of the winds passing by the port of Marseille arrive at the measuring station Marseille – Longchamp,
- fuel maritime regulations such as a change of policy in the fuel used,
- as well as an international lockdown during Spring 2020,

All these reasons could play a role on the fingerprint of the emissions of the maritime traffic as well as their intensities.

Based on the results presented here and those of the PAREA's campaign, the shipping source has been identified at our measurement site. But without results from near source measurement it would have been challenging to link this profile to a shipping source. Due to the particularity of the fingerprint the ratio between toluene and ethylbenzene or xylenes could be a promising ratio to the identification of the shipping impact on the concentration of NMHC in an urban atmosphere but further studies are needed to give a ratio value. Furthermore, in Marseille the ships are mainly using Marine Gas Oil (MGO) which is not necessary the case in other port cities. Also, the use of a ratio on aromatics could be tricky in cities where two wheels are more used than in the surrounding suburban area. Salameh et al., 2019 has showed that an increase of the two wheels proportion in the traffic fleet result to an enrichment of toluene, ethylbenzene and xylenes compared to benzene. Another issue to consider is that many ratio on aromatics are already used to identify sources in urban areas like benzene to toluene ratio for traffic exhaust, industry, coal combustion, biomass burning or aged air masses and m,p-xylene to ethylbenzene ratio is used to identify if the emissions are local or not (Liu et al., 2024).

### 3.3.6 Biogenic

The chemical profile presented in Fig. 11 for only both summers (2019 and 2020) is characterized by the presence of biogenic tracers like isoprene, limonene (sometimes co-eluted with 1,2,3-trimethylbenzene), and β-pinene. Between 65 % and 95 % of isoprene and limonene concentrations in summer are explained by this factor and between 40 % and 50 % for the β-pinene. These compounds are well known to be mainly emitted by biogenic sources (Guenther et al., 2012; Sindelarova et al., 2014).



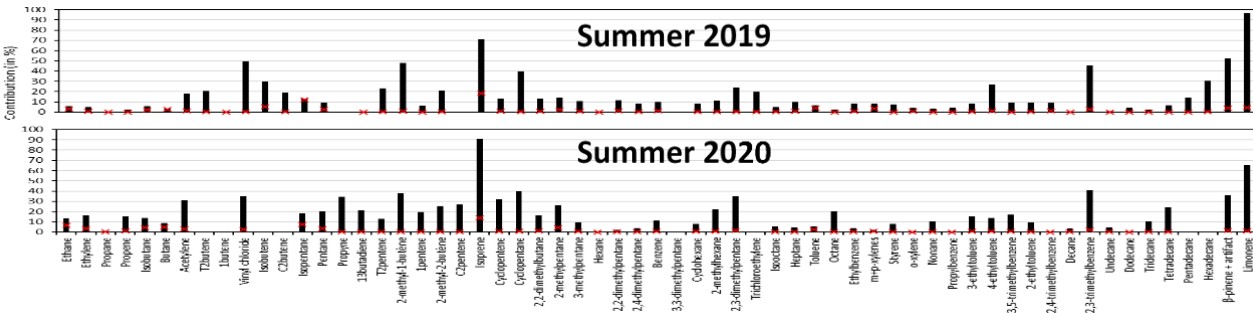

**Figure 11: Biogenic factor profile during the whole campaign. The black bars are the contribution of the factor to the species concentration and red crosses are the contribution of the species to the factor.**

Since our measurement site is located in the surrounding of the Longchamp Park composed of trees and plants, biogenic

emissions may impact the measurement. The diurnal profile of the factor's contribution (Fig. 13) shows a low background level during nighttime and an increase of emissions starting early in the morning 05:00 am UTC until a maximum at the beginning of the afternoon and a decrease after 04:00 pm UTC. This profile is similar to the temperature diurnal profile which is similar to the solar radiation profile (see Fig. S8 in supplement). The link between isoprene and terpenes with temperature is well known in the literature (Rasulov et al., 2010; Tingey et al., 1980). Therefore, this factor has been

identified as a biogenic factor.

High contribution of the biogenic factor occurs from southwest (sea breeze condition) to southeast wind corresponding to the location of the Longchamp Park beside the site. The pollution rose of this factor for summer 2019 is shown in Fig. 12.

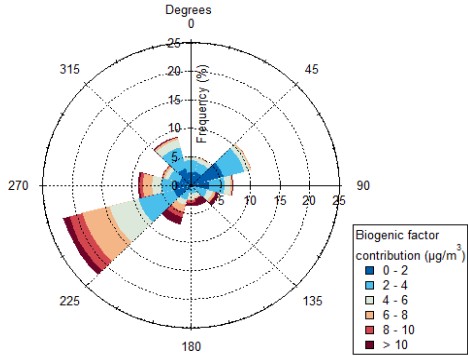

**Figure 12: Pollution rose of the biogenic factor during summer 2019.**

**3.3.7 IVOC**

The called IVOC factor is observed during all seasons and is characterized by the presence of C11 to C16 alkanes (see Fig. S9 in supplement). Their contribution to the factor composition profile varies from 20 % to 80 %. This disparity is explained



by the very low concentration of these compounds and therefore a high associated uncertainty. IVOC levels are generally below the C2-C10 NMHC levels (Fraser et al., 1997). Furthermore, only n-alkanes until hexadecane were measured during
this campaign and some studies showed that n-alkanes IVOC are not the major contributors to total IVOC which are highly dominated by an unresolved complex mixture of branched and cyclo alkanes and aromatics (Zhao et al., 2014, 2016). These compounds are mainly emitted by fossil fuel combustion (Jathar et al., 2017) and biomass burning (Hatch et al., 2018) and the importance of their measurements during field campaigns has already been demonstrated (Ait-Helal et al., 2014; Lu et al., 2020).

From fall 2019 to summer 2020 a proportion of around 20 % of ethane is explained by this factor, especially in fall 2019 and winter 2020 where the factor is driven by ethane rather than IVOC.

This factor shows no diurnal profile during Mistral events unlike during sea/land breeze conditions. During land breeze conditions this factor shows higher levels of IVOC (see part 3.1 for explanation) whereas during Mistral event, the IVOC levels are close to the ones observed during sea breeze phenomenon. This difference between nighttime Mistral event
concentrations and land breeze concentrations may be due to the wind speed difference between Mistral and land breeze. High wind speed during Mistral event helps the pollutants dispersion (Drobinski et al., 2007).

The Table 9 presents the R and the p-value of this factor with NO, $NO_2$, evaporation due to traffic factor, traffic exhaust factor and residential heating factor. A clear relationship is visible between the IVOC factor and $NO_x$ and factors linked to combustion processes during summers and winter 2020. This means that this factor is potentially linked to a combustion
process. Nevertheless, this correlation is not present during fall and spring seasons with the exception of spring 2019.







**Table 9: Pearson correlation coefficient and p-value for the IVOC factor with NO, NO₂, fuel evaporation factor, traffic exhaust factor and residential heating factor for all seasons. *** means a p-value < 0.001 %, ** a p-value between 0.001 % and 1 % and * a p-value >1 %.**

|  |  | Spring 2019 | Summer 2019 | Fall 2019 | Winter 2020 | Spring 2020 | Summer 2020 |
|---|---|---|---|---|---|---|---|
| $NO_2$ | **Pearson correlation coefficient** | 0.62 | 0.30 | 0.04 | 0.44 | -0.03 | 0.20 |
|  | **p-value (%)** | *** | *** | * | *** | / | *** |
| NO | **Pearson correlation coefficient** | 0.18 | 0.07 | -0.02 | 0.36 | -0.02 | 0.10 |
|  | **p-value (%)** | *** | * | / | *** | / | ** |
| Fuel evaporation | **Pearson correlation coefficient** | 0.54 | 0.29 | -0.02 | 0.37 | -0.11 | 0.24 |
|  | **p-value (%)** | *** | *** | / | *** | / | ** |
| Traffic exhaust | **Pearson correlation coefficient** | 0.40 | 0.38 | 0.16 | 0.47 | -0.03 | 0.18 |
|  | **p-value (%)** | *** | *** | *** | *** | / | *** |
| Residential heating | **Pearson correlation coefficient** | / | / | -0.14 | 0.31 | 0.02 | / |
|  | **p-value (%)** | / | / | / | *** | * | / |

The correlation coefficient between the IVOC factor and the traffic exhaust factor is the highest during winter 2020 which may indicate the link between IVOC factor and traffic exhaust during cold period. The cold start which leads to the increased contribution of traffic exhaust in winter is well known to increase the emission of IVOC as well and in particular from diesel engines (Drozd et al., 2019; Pereira et al., 2018) which represent 71,5 % of the car fleet in 2019 in Marseille according to AtmoSud. Furthermore, the importance of IVOC from evaporative sources like unburned diesel to the SOA formation has been explained by (Drozd et al., 2021) and are temperature and wind-speed dependent explaining the correlation of IVOC with the fuel evaporation from traffic as well as from traffic exhaust (table 9).

To summarize, there is a clear relationship between the IVOC factor and combustion processes but it is difficult to distinguish a specific combustion process especially that the chemical processes have an impact on the partitioning of IVOC between the particulate and gaseous phase.





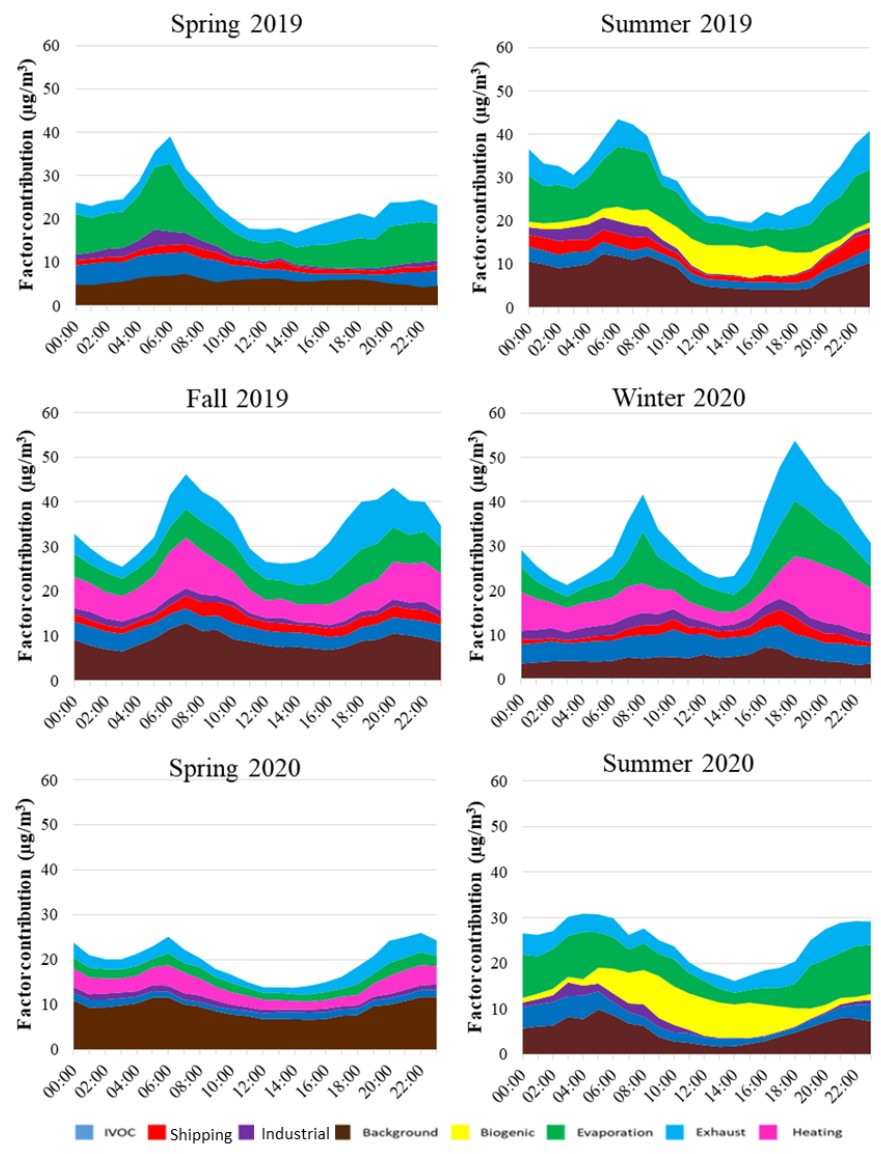

**Figure 13: Diurnal variation of each profile for each season. Hours are in UTC.**

### 3.3.8 Local and regional urban background

A factor attributed to the urban background is observed during the whole campaign and explains from 20 % to 40 % of the
variability of ethane over the period fall 2019 to summer 2020, and up to 60 % and 80 % in spring 2019 and summer 2019
respectively (Fig. S10).



The presence of other compounds like vinyl chloride and acetylene during some seasons could be due to specific local activities not well identified and separated by this PMF analysis. Globally, the fingerprint of this factor is characterized by long-lived compounds as already observed by many source apportionment studies (Salameh et al., 2016; Sauvage et al., 2009). The diurnal profile of the background factor shows a quite steady level along the day (Fig. 13).

To distinguish between the local and regional origin of this source, three days back trajectories have been modeled by the National Oceanic and Atmospheric Administration (NOAA) HYSPLIT tool for each hour. Then, the Potential Source Contribution Function (PSCF) (Ashbaugh et al., 1985) which is a commonly used function to determine the potential geographical origin of a source (Ara Begum et al., 2005 and references therein) has been applied. As shown in Fig. 14, many urban areas are pointed out as potential source areas contributing to this background factor. In the case of spring 2019, this source has for geographical origin an area in the east of the site corresponding to the Pô valley which is an area with important industrial activities in Italy, whereas in spring 2020, Nice located in the east of Marseille's site has been identified. In both situations, there is also a local origin to this factor, therefore it is identified as local and regional urban background.

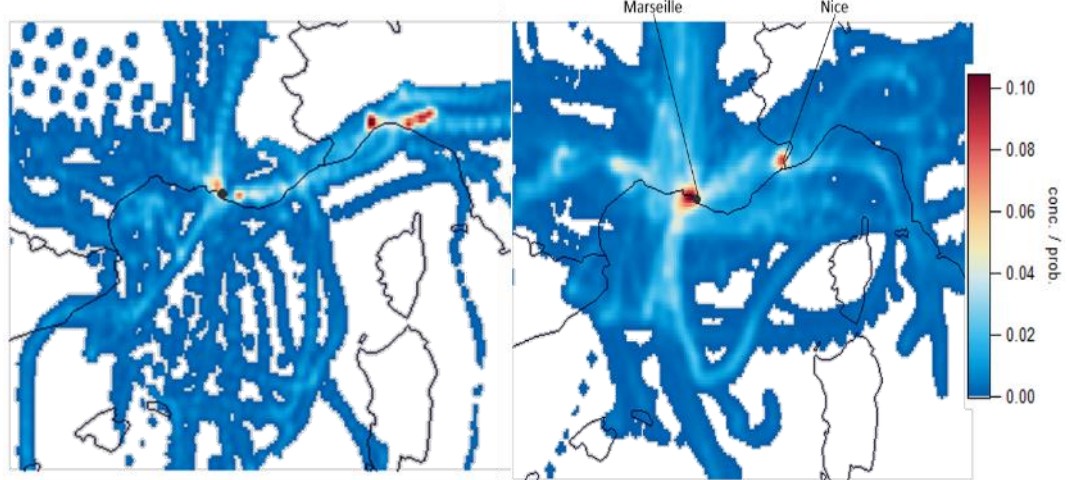

**Figure 14: PSCF for background factor contribution in summer 2019 (left) and spring 2020 (right).**

### 3.4 Factors contribution to the atmospheric pollution

The seasonal contributions of the PMF factors is presented in Fig. 15 in terms of (a) NMHC concentration, (b) $O_3$ formation potential (OFP) and (c) SOA formation potential (SOAFP). The OFP has been determined for each factor following the updated Maximum Incremental Reactivity (MIR) (Venecek et al., 2018) and the SOAFP following the SOA Potential (SOAP) determined by Derwent et al., 2010. Nevertheless, the MIR and SOAP correspond to the maxima of secondary pollutants formation potential which occur on summertime conditions when the photochemistry is the most active. Applying



the MIR and SOAP on wintertime data may lead to an overestimation of OFP and SOAFP especially for the most reactive species like alkenes. Therefore, wintertime OFP and SOAFP results have to be taken with caution.

Furthermore, this campaign being performed on an urban background measurement site and not near emission sources, the
results given in the fig. 15 are not the total OFP and SOAFP of emission sources but the $O_3$ and SOA that can still be produced by these emission sources when the pollutants arrive to the measurement site even if the impact of the photochemistry on our measured concentrations is negligible according to the observations made previously (see part 3.2).

Since alkenes are not considered, the OFP determined for spring 2020 is partially estimated. Concerning the SOAFP, it is probably underestimated for IVOC and biogenic compounds due to the absence of SOAP from the literature for limonene
and C13 – C16 alkanes, despite the fact that IVOC are large contributors to the SOA formation (Giani et al., 2019; Miao et al., 2021; Tkacik et al., 2012). Finally, the IVOC factor's contribution to NMHC concentration is likely overestimated due to the difficulty to separate the ethane from the urban background factor.

As for NMHC concentrations, traffic (evaporation and exhaust) is always the first contributor to OFP and SOAFP with a relative contribution varying from 30 % to 50 % for both OFP and SOAFP. Between evaporation and exhaust, the exhaust
part is the principal contributor to OFP and SOAFP due to the presence of alkenes which have a high OFP and aromatics which have a high SOAFP in this factor ($k_{OH}$[toluene] = 6.2x10$^{-12}$ molec.cm$^{-3}$.s$^{-1}$, $k_{OH}$[o-xylene] = 14.7x10$^{-12}$ molec.cm$^{-3}$.s$^{-1}$ (Atkinson, 1986)) whereas the evaporation factor is mainly composed of C4 – C5 alkanes, compounds with long lifetime ($k_{OH}$[butane] = 2.4x10$^{-12}$ molec.cm$^{-3}$.s$^{-1}$, $k_{OH}$[pentane] = 3.8x10$^{-12}$ molec.cm$^{-3}$.s$^{-1}$ (Atkinson and Arey, 2003)). The traffic is commonly the first emitter of NMHC in urban areas dealing with NMHC measurements (Panopoulou et al., 2021; Salameh
et al., 2016) but not when Oxygenated VOC are included (Baudic et al., 2016; Languille et al., 2020). This points out the importance to include OVOC in future studies since these pollutants are among the most emitted in the urban air (McDonald et al., 2018).





**Figure 15: Relative contribution of each factor to (a) the NMHC concentration, (b) the O₃ formation potential and (c) the SOA formation potential for each season.**





During the cold period, residential heating contributes significantly to NMHC levels with a mean contribution close to 7 µg.m$^{-3}$ which correspond to 20 % of the NMHC levels which is quite similar to the contribution of residential heating observed in other studies (Baudic et al., 2016; Panopoulou et al., 2021; Sauvage et al., 2009) and is considered as the second

contributor to NMHC concentrations after traffic. As described in part 3.3.3, the residential heating factor is composed of light alkanes but also alkenes (a high contributor of O$_3$ formation) and some aromatic compounds (a high contributor of SOA formation). Then, due to the high contribution to the OFP and SOAFP of some compounds present in this factor on one side and the seasonal variability of this factor, present only during wintertime on the other side, the contribution of residential heating to OFP and SOAFP according to the Fig. 15 might be overestimated.

According to the Fig. 15, the lowest contributors to NMHC concentrations are the industrial and shipping factors. Their contribution to OFP and SOAFP is totally different which is related to their chemical fingerprints. The industrial factor is mainly composed of isooctane, cyclohexane and benzene which are exclusively species with a low reactivity whereas the shipping factor is mainly composed of ethylbenzene and xylenes which are great contributors of SOAFP. Then despite a similar contribution to NMHC concentrations, the shipping factor is expected to contribute much more to the SOAFP than

the industrial factor.

### 3.5 Impact of Covid-19 lockdown on sources contribution

The year 2020 has been marked by the sanitary crisis leading to a lockdown all over France from 17$^{th}$ of March to 10$^{th}$ of May 2020. It is of interest to assess the impact of the lockdown on the VOC source apportionment. For that purpose, meteorological conditions during this period have to be firstly compared with the ones during the same period in 2019 (Table

555 10).

**Table 10: Meteorological conditions during lockdown period in 2020 and the similar period in 2019. Concerning wind speed only speed higher than 0.5 m.s$^{-1}$ are considered and concerning precipitation only hours with at least 1 mm of precipitation are considered.**

| | Minimal temperature (°C) | Maximal temperature (°C) | Mean temperature (°C) | Minimal wind speed (m.s$^{-1}$) | Maximal wind speed (m.s$^{-1}$) | Mean wind speed (m.s$^{-1}$) | Number of hours under Mistral event | Height of precipitation (mm) | Number of rainy hours |
|---|---|---|---|---|---|---|---|---|---|
| 17/03/19 to 10/05/19 | 6,3 | 23,2 | 13,7 | 0,5 | 3,8 | 1,2 | 267 | 101 | 31 |
| 17/03/20 to 10/05/20 | 3,8 | 24,8 | 15,0 | 0,5 | 2,8 | 0,9 | 179 | 26 | 17 |



The temperature range as well as the wind speed range during the lockdown in 2020 were in a similar range as in 2019. But when looking at the number of hours under Mistral events, there is a higher number of hours under Mistral event in 2019 in comparison to the same period in 2020. The Mistral is well known to be associated with low level of pollutants due to their dispersion away from their emission sources (Drobinski et al., 2007). Therefore, wind conditions during lockdown in 2020 were more favorable to the accumulation of pollutants than in 2019.

Finally, the global precipitation in this period and the number of hours with precipitation were lower in 2020 than in 2019 leading to an increase of the pollutant's concentrations due to the washing effect of precipitation on air pollution.

Briefly, during the lockdown period, meteorological conditions were favorable to the accumulation of pollutants in comparison to the same period in 2019 due to more stagnant and dryer air masses.

As shown in Table 11, besides the background factor, the contribution of all the factors are significantly lower in spring 2020

compared to spring 2019 even compared to the other seasons. Even if C2 – C6 alkenes measurements were invalidated during spring 2020, their absence does not explain the high decrease of the traffic exhaust factor's contribution. From spring 2019 to winter 2020, the percentage of C2 – C6 alkenes on the total measured NMHC concentrations from traffic exhaust was between 9.43 % and 16.75 % with a mean percentage of 12.98 %. Assuming a contribution of C2 – C6 alkenes equal to this mean for spring 2020, the total traffic contribution reaches 5.57 µg.m$^{-3}$ which is still about 50 % less than during other

seasons. This result is in good agreement with the decrease of $NO_x$ levels reaching 50 % compared to the levels measured at the same period during previous years (AtmoSud, 2020).

**Table 11: Contribution of each factor to NMHC emissions for all seasons. Results are given in µg.m$^{-3}$.**

| | IVOC | Fuel evaporation | Traffic exhaust | Total (traffic) | Industrial | Shipping | Residential Heating | Biogenic | Local and regional urban Background |
|---|---|---|---|---|---|---|---|---|---|
| **Spring 2019** | 3.40 | 7.81 | 4.01 | 11.82 | 1.29 | 1.35 | / | / | 5.71 |
| **Summer 2019** | 2.28 | 8.39 | 4.25 | 12.64 | 1.36 | 2.30 | / | 3.75 | 7.98 |
| **Fall 2019** | 3.50 | 5.74 | 6.01 | 11.75 | 1.32 | 2.10 | 6.79 | / | 8.91 |
| **Winter 2020** | 4.63 | 6.33 | 6.24 | 12.57 | 2.16 | 1.74 | 6.89 | / | 4.62 |
| **Spring 2020** | 1.42 | 2.13 | 2.72 | 4.85 | 1.04 | / | 3.22 | / | 9.14 |
| **Summer 2020** | 2.65 | 6.82 | 4.06 | 10.88 | 1.09 | / | / | 4.59 | 5.42 |



VOC concentrations are known to decrease since three decades (Waked et al., 2016) but because of the Covid-19 lockdown
the comparison between both springs and summers is not in the same condition and a longer study should be useful to
evaluate the inter-annual variability of VOC concentrations in Marseille.

## 4 Conclusions

A one-year-and-half field campaign for the measurement of 70 NMHC from C2 to C16 including IVOC has been conducted
from March 2019 to August 2020 at an urban background site in Marseille resulting in a unique high-quality hourly dataset
contributing to a better understanding of the atmospheric pollution in the Marseille area.

The EPA PMF 5.0 has been applied in order to identify the sources and quantify their contribution to NMHC concentrations.
During all the seasons, we identified six NMHC sources namely traffic exhaust, fuel evaporation, urban local and regional
background, industrial, shipping and an IVOC specific profile. Two seasonal sources have been identified such as the
biogenic source in summer and the residential heating in winter. There is a difference between cold and warm periods in the
contribution of traffic exhaust, which is 33% (2.0 µg/m$^3$) lower in warm periods compared to the cold periods.

The traffic related sources (exhaust and fuel evaporation) is the first contributor to NMHC concentrations measured by
contributing to 40 to 50 %, as well as to OFP and SOAFP during the whole campaign. In summer 2019 these sources are
contributing to 30 – 35 % of OFP and SOAFP. During wintertime, residential heating is the second contributor to NMHC
measured concentrations by contributing to 20 % of measured NMHC.

The weakest factors in terms of contribution to NMHC concentrations are the industrial and shipping factors as they have a
low background level with high episodic peaks. The shipping factor is mainly composed of ethylbenzene and xylenes
contributing to an important SOAFP due to the reactivity of these compounds, whereas the industrial factor is essentially
composed of benzene, isooctane and cyclohexane which have a low reactivity and therefore a low impact on OFP and
SOAFP.

Taking advantage of this unique datasets, we've been able to assess the effect of the lockdown due to the Covid-19 between
March 2020 and May 2020 in Marseille. A significant decrease of the factors' contribution has been shown for all factors
except the local and regional urban background, reaching 50 % for traffic related sources (exhaust and fuel evaporation).

A comparison of these sources' contribution with the local and regional emission inventories should be useful to evaluate
their accuracy for a better understanding of the atmospheric pollution occurring at Marseille. This comparison is even critical
since the traffic is contributing to 4 % of NMVOC emission in Marseille in 2019 according to the AtmoSud inventory which
is 10 times lower than our PMF analysis. However, studies including OVOC measurements shows a better agreement
between field measurements and inventory concerning the contribution of traffic to VOC concentrations. The presence of

OVOC measurement is then crucial to evaluate the local and regional emission inventories as well as helping to identify a source related to solvent use which is becoming an important VOC contributor in emission inventories.

Additionally, the measurement of IVOC is very challenging due to the absence of standard for these compounds, their very low concentrations in the atmosphere and the difficulty to separate most of them to individual species. Still, these compounds are of interest due to their contribution to SOA formation and their measurement is important in urban areas. Finally, if this study gives information about NMHC for 18 months the VOC concentration is decreasing since the last three decades and particularly when considering VOC emitted by traffic. Long-term measurements would be of interest to follow

the evolution of VOC concentrations.

**Data availability:** VOC data has been submitted to the EBAS database (https://ebas.nilu.no/).

**Author contribution**

SS and TS planned the campaign. MD and TL performed the campaign with the help of GG and AA. MD analyzed the data with the supervision of SS and TS. MD wrote the manuscript with the support and review from SS and TS.

**Competing interests**

The authors declare that they do not have any actual or potential financial and personal conflict of interests with other people or organizations.

**Acknowledgements**

This work was supported by the French Agency for Ecological Transition (ADEME) and the Mines-Telecom Institute Nord-
Europe (IMT Nord-Europe). We kindly acknowledge the INSU and the MISTRALS program for the financial support of the campaign. We also kindly acknowledge Ludovic Lanzi for his technical support throughout the field campaign. This study is a contribution to the RI-URBANS project (Research Infrastructures Services Reinforcing Air Quality Monitoring Capacities in European Urban and Industrial Areas, European Union's Horizon 2020 research and innovation program, Green Deal, European Commission, contract 101036245).



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
