# Peer review of "VOC sources and impacts at an urban Mediterranean area (Marseille – France)"

_EGUsphere, 2024_

## Author Response (AR1)

**Reviewer #1:**

As the authors mention, this is a unique and high-quality dataset describing VOCs in Marseille. The study features a well-executed source apportionment analysis using PMF modeling and provides valuable estimates of the impacts of these VOCs on ozone and secondary organic aerosol (SOA) formation. However, the manuscript would benefit from greater clarity regarding especially the new findings and a broader perspective. Additionally, some restructuring is recommended.

The results are currently discussed from a very localized perspective. It would strengthen the manuscript to explore the broader applicability of the methods and findings to other regions. For instance, the authors could discuss what implications the findings have for VOCs, OFP and SOAP in Europe more generally. The study could also highlight how a long-term dataset enables the investigation of seasonal variations in VOC sources, which are shown to be significant in this study, and compare these seasonal trends with emission inventories for Europe. How do the source contributions in this study relate to findings from other European locations or globally? What aspects of the emissions are unique to the Marseille site? Are there other areas in the world with similar unique emissions? Additionally, how does the source apportionment differ from studies conducted in Chinese megacities or the United States?

While source apportionment studies on VOCs in Europe have been conducted for over two decades, it would be helpful to explicitly outline what new insights this study provides and to place greater emphasis on these novel findings when presenting results and in the discussion.

The manuscript currently includes an extensive set of figures and tables. Please consider which are most critical and whether some could be presented more compactly or moved to the supplementary material. Figures should highlight new and significant results, while data supporting expected outcomes could be relocated to the supplement. Additionally, some figures could be presented in a clearer format.

*First, we would like to thanks the reviewer for accepting to review our work and providing all these valuable comments which were precious to enhance our article and to improve the comparison with other studies by pointing out the new insights of our article. To answer point by point:*

*Concerning VOCs, OFP, and SOAFP in Europe, there is generally a lack of recent studies. A study by Holland et al., 2023 showed that in London, the OFP from NMHC related to traffic sources has decreased, whereas in previous decades, traffic-related NMHC were by far the dominant contributors. This shift is explained by the numerous pollution control strategies implemented over the past 30 years and the fact that ethane and propane, which are related to non-traffic combustion sources have not been significantly affected by legislation (Holland et al., 2023).*

*But a study by In 'T Veld et al., 2024 found that traffic-related sources remained the primary contributors to OFP and SOAFP in Barcelona, based on measurements taken at an urban background site in 2022. This study also collected data from a regional background site, showing that in this location, biogenic sources were the main contributors to OFP and SOAFP. Interestingly, our study presents the same findings as those reported by In 'T Veld et al., 2024.*

*The top three contributors to OFP over the entire campaign were m,p-xylenes (14.22%), toluene (10.61%), and ethene (6.59%), compared to ethene (20.47%), ethane (15.13%), and n-butane (12.17%) in London Eltham, an urban background site in the UK (Holland et al., 2023). It is important to note that for the London Eltham data, the Photochemical Ozone Creation Potential (POCP) was used instead of the Maximum Incremental Reactivity (MIR) method applied in our study. However, it has already been shown that both methods yield comparable results (Derwent et al., 2010). In our study, aromatics played a more significant role in OFP compared to the findings of Holland et al., 2023.*

*The following sentences were added on line 528:*

*"The same observation was made in a recent study conducted at an urban background site in Barcelona (In 'T Veld et al., 2024). In contrast, a recent study by Holland et al., 2023 revealed that the top contributors to OFP among NMHCs were ethene, ethane and propane, based on three decades of measurements at both a traffic-related site and an urban background site in London. It is important to note that this study did not evaluate OFP using the MIR method but rather the Photochemical Ozone Creation Potential (POCP). However, previous research has shown that both methods produce comparable results (Derwent et al., 2010). More recent studies using long-term datasets in urban background areas across Europe would be valuable to determine which results—those of Holland et al., 2023 or those of our study—are more representative."*

*Concerning the importance of a long-term dataset to investigate the seasonal variations of VOC sources it is addressed as a perspective at the end of the conclusion (lines 613-615). With 18 months we cannot conclude on a season if it is representative or particular season with particular concentrations or if it is globally what is observed each year at this place. As shown by Debevec et al., 2021, interannual variability must be considered, as it can be influenced by different air masses. This observation was also highlighted by Putaud et al., 2021, who assessed the impact of meteorology on pollutant concentrations by comparing data from March and April 2020 with the same period in previous years to evaluate the effect of lockdown measures. Moreover, as indicated in lines 613–615, VOC concentrations have decreased in France over the past three decades. Therefore, long-term measurements are essential to accurately track and verify VOC concentration trends.*

*Concerning the comparison with seasonal trends observed in European emission inventories, we compared our results with both the Copernicus Atmospheric Monitoring Service (CAMS) and the Community Emissions Data System (CEDS), which are emission inventories (EI) with a monthly temporal resolution (results not shown here). The considered emission fluxes corresponded to an area covering the city of Marseille. The analysis was conducted for total VOC emissions, emissions from road transport, and emissions from residential sources. It is important to note that the residential category in the EI includes not only heating but also cooking and boilers, although heating is considered the dominant source. For the year 2019, total VOC emissions were more than 30% higher in winter than in summer for both EI datasets (+39% for CAMS and +35% for CEDS). This result aligns with our observations in Figure 4, where concentrations were approximately 20% higher in winter than in summer. For road transport, no significant seasonal differences were observed in either EI, which is consistent with our findings in Table 11, where seasonal variations in the total traffic contribution to VOC concentrations were not substantial. Regarding residential emissions, they were higher in winter than in summer by 82% according to CAMS and 46% according to*

*CEDS. Once again, this is in agreement with our observations, as we were able to identify the residential heating source only during the winter period.*

*Concerning the comparison with other sites in Europe and Worldwide, the aspects of the emissions that are unique to the Marseille site, the areas with similar unique emissions and how the PMF studies differ with other studies worldwide, a new table is added doing a listing of PMF results in Marseille (this study), Paris (Baudic et al., 2016), Athens (Panopoulou et al., 2021), Beirut (Salameh et al., 2016), Beijing (Cui et al., 2022) and Denver (Frischmon and Hannigan, 2024) and the following part is added right after the identification of the part 3.4.8 Local and regional urban background (which is noted as part 3.3.8 due to a mistake on the numbering of the parts).*

*"The PMF results obtained in this study were compared with those from other cities in France (Paris, Baudic et al., 2016), Europe (Athens, Panopoulou et al., 2021), the eastern Mediterranean basin (Beirut, Salameh et al., 2016), China (Beijing, Cui et al., 2022), and the USA (Denver, Frischmon and Hannigan, 2024), using the same measurement technique as our study. These results are summarized in Table 10.*

*Traffic exhaust and fuel evaporation sources were identified in all these studies, although some studies distinguished these factors with greater precision. For instance, Salameh et al., 2016 were able to separate local and regional traffic exhaust emissions as well as fuel evaporation from both traffic and fuel storage facilities. Residential heating and local and regional background sources were also identified in most of these studies, but they were often mixed with other sources. In Athens, residential heating was combined with the local and regional background, while in Paris, the local and regional background was merged with the natural gas source.*

*The IVOC and shipping sources were only observed in Marseille. This can be explained by the fact that heavy n-alkanes were specifically measured in Marseille, and the site is located near the port. Finally, the industrial factor was also identified in Beijing and Denver, but its characteristics varied significantly across cities. As previously mentioned, the industrial factor in Marseille explains a significant portion of the measured cyclohexane and benzene concentrations and is linked to a plastic production industry located east of the station. In Beijing, the industrial factor included high concentrations of VOCs, with the most abundant being 2,3,4-trimethylpentane, n-butane, and methylcyclopentane. This factor was associated with petrochemical industries (Cui et al., 2022). In Denver, the industrial factor was attributed to a refinery and fuel storage tanks and was therefore defined as an oil extraction, processing, and evaporation factor (Frischmon and Hannigan, 2024).*

*Many recent studies present PMF results based on Proton Transfer Reaction-Mass Spectrometry (PTR-MS) measurements. PTR-MS has the advantage of detecting OVOCs and highly reactive species, but it does not measure alkanes or distinguish between isomers. Consequently, due to differences in the measured compounds, these studies often identify sources that cannot be detected using gas chromatography techniques alone. For example, Languille et al., 2020 identified solvent use in Paris, and Gkatzelis et al., 2021 showed that in US cities, Volatile Chemical Products (VCPs) are a major source of VOCs. Finally, Simon, 2023 identified cooking emissions by combining a PTR-MS dataset with an Aerosol Chemical Speciation Monitor (ACSM) dataset in Paris.*

*Splitting datasets by season during PMF analysis is common for aerosol studies but remains uncommon for VOC studies, where measurements are often conducted during specific periods of each season. Additionally, PMF studies based on PTR-MS measurements typically cover only a few weeks to a few months. One possible reason is the limitation of the EPA PMF model, which can handle a maximum of 500,000 data points (Hopke et al., 2023), posing a constraint for near-real-time instrumentation. As a result, the length of our dataset is a strength, allowing the coverage of multiple years of seasonal variations. Furthermore, IVOCs are still rarely measured, and very few PMF studies include these compounds. Finally, our campaign was already in progress when the 2020 lockdown occurred, providing valuable insights into its impact on VOC concentrations. All these differences from other VOC source apportionment studies highlight the richness of the information obtained through this analysis."*

| | *Marseille March 2019 – August 2020 (this study)* | *Paris 2010 yearlong (Baudic et al., 2016)* | *Athens March 2016 – February 2017 (Panopoulou et al., 2021)* | *Beirut 2 weeks during summer 2011 and winter 2012 (Salameh et al., 2016)* | *Beijing between 1 and 2 month each season in 2019 (Cui et al., 2022)* | *Denver many datasets of few weeks to a year between 2018 and 2022 (Frischmon and Hannigan, 2024)* |
|---|---|---|---|---|---|---|
| *Biogenic* | ✓ | ✓ | | | | |
| *Traffic exhaust* | ✓ | ✓ | ✓ | ✓ | ✓ | ✓ |
| *Fuel evaporation* | ✓ | ✓ | ✓ | ✓ | ✓ | ✓ |
| *Industrial* | ✓ | | | | ✓ | ✓ |
| *Shipping* | ✓ | | | | | |
| *Residential heating* | ✓ | ✓ | ✓ | | ✓ | |
| *IVOC* | ✓ | | | | | |
| *Local and regional background* | ✓ | ✓ | ✓ | ✓ | | |
| *Natural gas* | | ✓ | ✓ | | | ✓ |
| *Solvent use* | | ✓ | | | ✓ | |
| *Liquefied petroleum gas* | | | | ✓ | | ✓ |
| *Gas leakage* | | | | ✓ | | |

*Concerning the outlying of the new insights, the following sentences are added or modified to the manuscript:*

*Line 411:* *"furthermore this profile is not comparable to any profile found in source apportionment studies in the literature and so is a novelty here."*

*Line 449:* *"But if few studies are now including IVOC measurement, these compounds are still not considered on source apportionment analysis probably due to their low concentrations and their high associated uncertainties, and their partitioning between gas and particule phase. Then this PMF source offers new information to the scientific community."*

*Line 548-550:* *"Then despite a similar contribution to NMHC concentrations, the shipping factor is expected to contribute much more to the SOAFP than the industrial factor and is even the strongest potential contributor to the formation of SOA in summer 2019 above the road traffic. This result highlights the necessity to better characterize the shipping profile and our results will be helpful for further studies."*

*Finally, concerning the important number of figures and tables, the figures 5, 7, 8, 9, 11, S6, S9 and S10 are replaced by the following figure 5. The figure 14 and the tables 5, 6, 7, 9 and 10 are moved in the supplementary material. Finally, the figures 2, 10, 12 and S5 are merged into the following figure S4.*

[Figure]

*Figure 5:* *Contribution to species concentration of (a) traffic exhaust, (b) fuel evaporation from traffic, (c) residential heating, (d) industrial, (e) shipping, (f) biogenic, (g) IVOC and (h) local and regional urban background factors for each season. Results are given in percentage.*

[Figure]

**Figure S4:** *(a) wind rose, (b) pollution rose of the fuel evaporation from traffic factor, (c) pollution rose of the industrial factor and (d) pollution rose of the biogenic factor for the whole campaign.*

**Specific comments:**

1. **Section 2.2:** What type of inlet system was used?

*It was a 3-meter Sulfinert treated INOX line. A new sentence has been added in blue in the following text:*

*"The combination of the measurement of these two devices gives us the mixing ratio of 35 common compounds. The sampling line was a 3-meter Sulfinert treated INOX line."*

2. **Section 2.**2: Nafion dryers are known to introduce high and variable backgrounds for certain light alkenes, such as trans- and cis-butene. Was the blank measurement performed through the Nafion dryer?

*Yes, the blank was performed through the Nafion dryer. In our case the Nafion dryer had an impact on 1-butene and isobutene. We also had low artifacts for butanes and benzene on the TD-GC-2FID and these artifacts have been considered when determining the concentration of these compounds by subtracting their respective blanks values. Concerning the other compounds in both TD-GC-2FID and TD-GC-FID the blanks values were due to the inherent*

*variability of the noise and were below the LoD of the compounds. The sentence line 124 has been changed as follow:*

 *"Both instruments repeatability, reproducibility, and blank, have been checked many times during the campaign in the same conditions as ambient air sampling."*

3. **Figure 3:** Consider moving this figure to the supplementary material.

*The figure is now in the supplementary material and has been changed as follow:* **"Figure S4: Scatter plot of (a) ethene, (b) m,p-xylene and (c) n-pentane vs. benzene (in ppb) in winter 2020 (left) and summer 2019 (right) during daytime (red) and nighttime (blue)."**

*And the line 201 has been changed as follow:* *"Figure S4 shows the scatterplot of ethene, m,p-xylene and n-pentane with benzene."*

4. **Section 3.3:** It would be helpful to include the concentration levels of all studied compounds, perhaps in a table in the supplementary material.

*A new table was added in the supplementary material with the mean concentration, the median concentration and the percentage of values below the limit of detection. See below the new table:*

**Table S3:** *Mean and median concentration and number of measurements below the LoD for all the measured compounds during the whole campaign (the number of measurements below the LoD and the total number of measurements are given in the parenthesis).*

| Compounds | Mean concentration (ppt) | Median concentration (ppt) | Number of measurements below the LoD (%) |
|---|---|---|---|
| Ethane | 2278 | 1749 | 0 (2/9675) |
| Ethene | 830 | 535 | 3 (244/7177) |
| Propane | 1024 | 779 | 1 (53/9675) |
| Propene | 233 | 156 | 11 (783/7177) |
| Isobutane | 653 | 479 | 0 (8/9675) |
| Butane | 1259 | 906 | 0 (6/9675) |
| Acetylene | 483 | 347 | 17 (1329/7973) |
| Trans-2-Butene | 67 | 44 | 20 (1467/7177) |
| 1-Butene | 137 | 100 | 10 (696/7177) |
| Vinyl Chloride | 161 | 59 | 30 (2945/9675) |
| Isobutene | 311 | 267 | 5 (367/7177) |
| Cis-2-butene | 56 | 38 | 30 (2155/7177) |
| Neopentane | 13 | 12 | 96 (9267/9675) |
| Isopentane | 743 | 540 | 0 (6/9675) |
| Pentane | 331 | 248 | 0 (10/9417) |
| Propyne | 43 | 42 | 99 (9595/9675) |
| 1,3-butadiene | 62 | 22 | 63 (4538/7177) |
| 3-methylbutene | 14 | 11 | 90 (6491/7177) |
| Trans-2-pentene | 29 | 10 | 56 (4024/7177) |
| 2-methyl-1-butene | 38 | 11 | 53 (3836/7177) |
| 1-pentene | 23 | 11 | 76 (5482/7177) |
| 2-methyl-2-butene | 31 | 17 | 81 (5843/7177) |

| | | | |
|---|---|---|---|
| Cis-2-pentene | 17 | 9 | 76 (5438/7177) |
| Butyne | 12 | 9 | 93 (9036/9675) |
| Isoprene | 343 | 241 | 20 (1745/8734) |
| Cyclopentene | 65 | 27 | 59 (5289/8905) |
| Cyclopentane | 32 | 18 | 75 (6669/8905) |
| 2,2-dimethylbutane | 66 | 53 | 38 (3373/8905) |
| 2-methylpentane | 188 | 152 | 0 (0/8905) |
| 3-methylpentane | 83 | 59 | 1 (54/9826) |
| 1-Hexene | 13 | 4 | 73 (7200/9826) |
| Hexane | 97 | 69 | 3 (256/9826) |
| 2,2-dimethylpentane | 78 | 56 | 3 (317/9826) |
| 2,4-dimethylpentane | 48 | 34 | 8 (825/9826) |
| 2,2,3-dimethylbutane | 6 | 6 | 98 (9643/9826) |
| Benzene | 187 | 136 | 0 (3/9826) |
| 3,3-dimethylpentane | 15 | 6 | 91 (8979/9826) |
| Cyclohexane | 172 | 112 | 3 (265/9826) |
| 2-methylhexane | 57 | 36 | 23 (1529/6787) |
| 2,3-dimethylpentane | 31 | 15 | 62 (4213/6787) |
| Trichloroethylene | 9 | 6 | 100 (9821/9826) |
| Isooctane | 37 | 23 | 20 (1931/9826) |
| Heptane | 67 | 47 | 6 (586/9826) |
| Toluene | 506 | 370 | 0 (3/9826) |
| Octane | 17 | 8 | 66 (5701/8602) |
| Tetrachloroethylene | 168 | 165 | 99 (8538/8602) |
| Ethylbenzene | 95 | 69 | 0 (43/9826) |
| m,p-xylenes | 349 | 254 | 0 (18/9826) |
| Styrene | 10 | 6 | 39 (3839/9826) |
| o-xylene | 123 | 86 | 1 (60/9826) |
| Nonane | 23 | 17 | 5 (496/9826) |
| Isopropylbenzene | 5 | 4 | 90 (8854/9826) |
| α-pinene | 4 | 3 | 88 (8691/9826) |
| Propylbenzene | 12 | 6 | 39 (3818/9826) |
| 3-ethyltoluene | 39 | 28 | 22 (1901/8769) |
| 4-ethyltoluene | 53 | 39 | 15 (1298/8769) |
| 1,3,5-trimethylbenzene | 21 | 8 | 61 (5317/8769) |
| 2-ethyltoluene | 17 | 12 | 39 (3827/9826) |
| 1,2,4-trimethylbenzene | 92 | 68 | 4 (359/9826) |
| Decane | 38 | 27 | 5 (502/9826) |
| 1,2,3-trimethylbenzene | 45 | 29 | 11 (1093/9826) |
| Limonene | 11 | 3 | 59 (5845/9826) |
| Butylbenzene | 5 | 3 | 79 (7808/9826) |
| Undecane | 24 | 20 | 8 (753/9826) |
| Dodecane | 15 | 11 | 29 (2833/9826) |
| Tridecane | 9 | 6 | 42 (4112/9826) |
| Tetradecane | 7 | 2 | 58 (5694/9826) |
| Pentadecane | 3 | 2 | 83 (8142/9826) |
| Hexadecane | 3 | 2 | 90 (8865/9826) |

5. **Lines 222–223:** Could the lower VOC concentrations in Athens (and possibly Beirut) compared to Marseille be due to poorer ventilation in those cities?

*The VOC concentrations in Athens and Beirut are higher than in Marseille despite the lower wind speed measured at our measurement site and many reasons can explain this observation such as the period of the measurement campaign (for example, the campaign in Beirut took place only during summer and winter), the year of the studies, the intensities of the sources...*

6. **Table 8:** The term "Rheating/BCwb" is not explained here and may already be presented in Table 7. Please clarify.

*You're right, the "Rheating/BCwb" is already presented in Table 7. This line is now deleted of the table 8. The table 8 looks now as below:*

**Table 8: Benzene to Toluene ratio for the traffic factor and the residential heating factor from fall 2019 to spring 2020.**

|                     | Fall 2019 | Winter 2020 | Spring 2020 |
|---------------------|-----------|-------------|-------------|
| Residential heating | 0.44      | 0.94        | 0.84        |
| Traffic             | 0.14      | 0.12        | 0.11        |

7. **Lines 365–367 & related lines on terpenes:** The results on terpenes are likely not quantitative due to losses and isomerization caused by the Nafion dryer. Please include a comment in the manuscript addressing this limitation or reconsider detailed discussions of these compounds.

*As indicated line 92 the Nafion dryer was used for the TD-GC-2FID specifically. The TD-GC-FID which is the one measuring limonene did not have any Nafion dryer and is able to measure some terpenes. It is an upgraded version of the instrument used by Panopoulou et al., 2020 (*https://doi.org/10.1016/j.atmosenv.2020.117803*) in Athens (the upgrade allows us to measure the NMHCs until C16 against C12 in the version used in Athens). To avoid any confusions the following modifications were made:*

*Lines 92-95: "A nafion dryer has been added in the sampling line of the TD-GC-2FID specifically to remove water […] on the most reactive species like alkenes (Bourtsoukidis et al., 2019)."*

*Lines 366-367: "A comparison between limonene and benzene and toluene in winter measured by the TD-GC-FID shows a clear correlation specially during nighttime."*

8. **Lines 367–368:** Note that cleaning and personal care products are also significant sources of limonene.

*Indeed, even if we were not able to detect these sources during our analysis they cannot be discarded. Then this precision is added line 368 «It is also important to note that even if these sources were not identified in our studies limonene can also be emitted in the cities by other anthropogenic sources such as cleaning or personal care products (Borbon et al., 2024). Then a possible impact of these sources on the measured concentrations of limonene cannot be totally discarded"*

9. **Lines 483–485:** Higher alkanes could serve as tracers for indoor air (e.g., Mai et al., 2024; https://doi.org/10.1016/j.heha.2023.100087). Please consider referencing this study.

*The following sentence is added line 482 « It is also worth noting that higher alkanes can be at stronger concentrations in indoor air than in outdoor air (Mai et al., 2024) meaning a possible origin from residential sources when performing outdoor measurements."*

10. **Figure 14:** Consider moving this figure to the supplementary material.

*The figure 14 is now the figure S11 and the lines 499-500 were modified as indicated below:*

*"As shown in Fig. S11, many urban areas are pointed out as potential source areas contributing to this background factor."*

11. **Line 517:** Photochemistry remains critical for the most reactive VOCs. Please clarify this point.

*Indeed, even though Figure 3 indicates that photochemistry had a negligible impact on the measured VOC concentrations, the MIR values are derived for summer conditions. Additionally, alkene concentrations were higher in winter due to emissions from residential heating (Section 3.3.3 in the initially submitted draft, Section 3.4.3 in the revised draft). These factors contribute to a likely overestimation of OFP and SOAFP in winter. Consequently, the following sentence has been added at line 513: "In addition, the concentration of alkenes was higher in winter than in summer due to residential heating. This observation, combined with the application of MIR values, can lead to a significant overestimation of OFP and SOAFP in winter."*

12. **Lines 521–522 & 613–615:** These sentences are difficult to follow. Please rephrase for better clarity.

*these sentences were rephrased as follow:*

*Lines 521-522: "Finally, the IVOC factor's contribution to NMHC concentration is likely overestimated due to the difficulty to separate the ethane from the urban background factor." => "Finally, if the IVOC factor is well identified by the PMF analysis it is nevertheless difficult to correctly separate it from the urban background factor and specially from ethane. The consequence is an overestimated contribution of this factor to the NMHC concentration"*

*Lines 613-315: "Finally, if this study gives information about NMHC for 18 months the VOC concentration is decreasing since the last three decades and particularly when considering VOC emitted by traffic. Long-term measurements would be of interest to follow the evolution of VOC concentrations." => "Finally, if this study gives information about NMHC for 18 months a decreasing of VOC concentration is observed more globally in France since the last three decades and particularly when considering VOC emitted by traffic. Then long-term measurements would be of interest to follow the evolution of VOC concentrations. Furthermore, long-term datasets are also of interest to evaluate the interannual variability.".*

13. **Lockdown effects:** While the authors conclude that factor contributions decreased significantly during the lockdown, it would be valuable to quantify the impact on air quality. How strongly did the lockdown affect the OFP and SOAP of the studied VOCs?

.

*Unfortunately, as indicated on line 518, alkenes were not considered in spring 2020. Since they are strong contributors to OFP, we cannot assess the extent to which the lockdown affected OFP. Regarding SOAFP, spring 2020 exhibited lower SOAFP values compared to other seasons, except for spring 2019. Therefore, we cannot draw definitive conclusions about the impact of the lockdown on SOAFP.*

*The following sentences have been added at line 576:*

*"To estimate the impact of the lockdown on SOAFP, we examined the absolute values of SOAFP across seasons. However, it is important to note that these absolute values are highly uncertain, as they assume that all measured compounds react optimally to form SOA. Nonetheless, they provide insights into how seasonal VOC concentrations influence SOAFP. In addition, MIR and SOAP values determined under summer conditions were applied uniformly across all seasons, limiting the validity of seasonal comparisons. Across all seasons, SOAFP values ranged between 584 and 710 µg/m³, except for spring 2019 and spring 2020, where values were 343 and 392 µg/m³, respectively. Consequently, we cannot conclude whether the lockdown had a significant impact on SOAFP.*

*It is important to note that pollutants were affected differently during the lockdown. For instance, in Marseille, while NOx levels rapidly decreased by 50% compared to previous years, fine particle concentrations were higher between March 17 and April 17 (AtmoSud, 2020). Similarly, a study by Putaud et al. (2021) assessed the specific impact of the lockdown on pollutant concentrations around Milan, Italy. They found that $NO_2$ levels decreased by 30% in urban background areas, whereas $PM_{10}$ concentrations remained largely unchanged. This was attributed to the reduction in traffic-related PM emissions being counterbalanced by an increase in PM emissions from residential heating. Therefore, an SOAFP value that appears unaffected by the lockdown may not be unexpected."*

**Reviewer #2:**

The manuscript presents long-term VOC measurements at an urban site in Marseille using TD-GC-FID. The results are further analyzed with PMF for source apportionment across different seasons. Long-term measurements of speciated VOCs are rare and valuable globally, making this manuscript a significant contribution to understanding urban VOC sources.

However, the manuscript contains 15 figures and 12 tables, which might overwhelm readers. It would be helpful to streamline these details - consider consolidating some figures or moving others to the supplementary information. For instance, Figures 5, 8, 9, and 11 display multi-panel factor profiles across multiple seasons, which may be too detailed for a broad ACP audience. Simplifying these figures (perhaps like Figure 13, which is clear and informative) to show seasonal changes in profiles could reduce cognitive load for readers and enhance the presentation. Additionally, displaying a few key factor profiles in the main text, while relegating others to the SI, could balance detail with readability. Grouping the wind

roses of different factors together in one place for easier comparison would also improve clarity.

Overall, the manuscript is compelling, offering novel insights into regional VOC sources in the Mediterranean urban region of Marseille. Streamlining the presentation would improve coherence and accessibility.

*We would like to thanks the reviewer for their evaluation of our study. All their valuable comments allow us to go further in the analysis of our results.*

*Concerning the important quantity of figures and tables, the figures 5, 7, 8, 9, 11, S6, S9 and S10 are replaced by the figure 5 presented in the answer to the general comment of the first reviewer. The figure 14 and the tables 5, 6, 7, 9 and 10 are moved in the supplementary material. Finally, the figures 2, 10, 12 and S5 are merged into the figure S3 presented in the answer to the general comment of the first reviewer.*

**Specific Comments:**

1. **Comparisons with Other Studies**: Table 4 does a great job comparing VOC mixing ratios in Marseille with other Mediterranean cities. It would be helpful to emphasize what makes the Marseille region unique, as well as the similarities and differences with other cities. A brief discussion addressing the "So what?" factor - why these comparisons matter - would enhance the impact of the findings.

*. The following sentences were added line 222: "Marseille has the particularity to have the port inside the city and at 2 – 3 km from the measurement site. The measurement site in Beirut is a suburban site (6km south-east of Beirut downtown) and is surrounded by a forested pine area. The site was occasionally exposed to very high concentrations of i-pentanes and n-pentanes in northern wind conditions (more than 40 ppb). Finally, the Athens measurement site is in the center of Athens and is on the top of a hill. This city has the particularity to be highly polluted in VOCs in comparison with other Mediterranean cities and has been considered by Panopoulou et al., 2018."*

*Erratum: in this table an error was made on the values of isopentane for Beirut. For Paris median values instead of mean values in µg/m³ instead of part per billion were considered followed by some reading errors. These errors change one observation made lines 223 – 225 and the sentence has been corrected as follow "On the other hand, measured concentrations at Marseille are lower than those measured in other French cities (Paris, Lyon, Strasbourg) especially ethane, and ethylene and benzene". See below the corrected value. Also, based on the next remark, the number of significant has been changed on this table to be consistent in all the manuscript.*

| | Beirut winter 2012 (Salameh et al., 2015) | Beirut summer 2011 (Salameh et al., 2015) | Athens winter 2016 (Panopoulou et al., 2018) | Marseille winter 2020 (this study) | Marseille summer 2019 (this study) | Paris 2010 yearlong (Baudic et al., 2016) | Strasbourg yearlong (2003 – 2013) (Waked et al., 2016) | Lyon yearlong (2007 – 2013) (Waked et al., 2016) |
|---|---|---|---|---|---|---|---|---|
| **Ethane** |  2.78 |  1.55 | 4.50 | 2.40 | 2.10 |  3.64 |  2.95 | 4.09 |
| **Ethylene** |  2.05 |  3.30 | 4.10 | 1.11 | 0.61 |  1.33 |  1.80 |  2.79 |
| **Acetylene** |  2.13 |  2.24 | 4.20 | 0.70 | 0.59 |  0.63 | 0.85 | 0.84 |
| **Isopentane** |  2.31 |  4.02 | 4.70 | 0.60 | 1.10 |  0.75 | 0.7 | 1.06 |
| **Benzene** |  0.53 |  0.61 | 0.80 | 0.60 | 0.14 |  0.32 |  0.32 | 0.47 |
| **Toluene** |  2.11 |  3.80 | 2.20 | 0.50 | 0.54 |  0.82 | 0.3 | 1.43 |
| **Ethylbenzene** |  0.26 |  0.52 | 0.40 | 0.14 | 0.10 | / | / | / |
| **m,p-xylenes** |  0.87 |  1.78 | 1.20 | 0.30 |  0.39 | / | / | / |
| **o-xylene** |  0.31 |  0.63 | 0.40 | 0.10 | 0.14 | / | / | / |

2. **Inconsistent Decimal Usage**: I noticed inconsistent use of decimal points. In some tables, dots are used, while commas are used in most figures. This inconsistency could confuse readers, particularly American audiences, who might misinterpret a comma as a thousands separator. For example, Figure 3 could be misread as showing very high concentrations in ppm when the authors likely intended for the comma to indicate decimal points. I recommend standardizing the decimal notation (either dots or commas) throughout the manuscript. Also, consider reducing the number of significant figures in some cases - 10,000 ppb of toluene could be expressed as 10.0 or 10.00 ppb for clarity.

*We have corrected the figures 3 (now figure S4) and 4 (now figure 2) and the table 10 (now table S10) with dots instead of comas and we have modified every figures and tables with VOC concentrations to have a harmonized number of significant between them. This number of significant was set to two. See below the corrected figures S4 and 2 and the table S10.*

[Figure]

**Figure S4: Scatter plot of (a) ethene, (b) m,p-xylenes and (c) n-pentane vs. benzene (in ppb) in winter 2020 (left) and summer 2019 (right) during daytime (red) and nighttime (blue).**

[Figure]

**Figure 2: Monthly variability of NMHC families measured during the one year and half campaign.**

**Table S10: Meteorological conditions during lockdown period in 2020 and the similar period in 2019. Concerning wind speed only speed higher than 0.5 m.s⁻¹ are considered and concerning precipitation only hours with at least 1 mm of precipitation are considered.**

|  | Minimal temperature (°C) | Maximal temperature (°C) | Mean temperature (°C) | Minimal wind speed (m.s⁻¹) | Maximal wind speed (m.s⁻¹) | Mean wind speed (m.s⁻¹) | Number of hours under Mistral event | Height of precipitation (mm) | Number of rainy hours |
|---|---|---|---|---|---|---|---|---|---|
| 17/03/19 to 10/05/19 | 6.3 | 23.2 | 13.7 | 0.5 | 3.8 | 1.2 | 267 | 101 | 31 |
| 17/03/20 to 10/05/20 | 3.8 | 24.8 | 15.0 | 0.5 | 2.8 | 0.9 | 179 | 26 | 17 |

3. **Mistral Impact**: Section 3.3.7 discusses the impact of Mistral events. Is the hypothesis that these events dilute the sources or that they alter air circulation patterns in a way that affects VOC concentrations? Clarifying this point would strengthen the argument.

*The Mistral event is known to going from the Northern land until many kilometers after the Mediterranean coasts. The hypothesis is that the VOCs are advected away from their sources of emissions over the Mediterranean Sea. See the precision added to the following sentence lines 454 – 456.*

*"This difference between nighttime Mistral event concentrations and land breeze concentrations may be due to the wind speed difference between Mistral and land breeze. High wind speed during Mistral event contributes to the pollutants advection from their sources to the Mediterranean Sea (Drobinski et al., 2007)."*

4. **VOC-NOx Correlations**: The relatively low Pearson coefficients in Table 9 for VOC-NOx correlations are intriguing. Have the authors considered constraining the analysis to daytime hours or using different subsets of VOCs to explore potential stronger dependencies? Also, I noticed that ozone was not included in the correlational analysis - was there a particular reason for this omission?

*The Pearson coefficient presented here concerns only the linear alkanes from n-undecane to n-hexadecane (referred to as IVOCs in this study). Firstly, these IVOCs were present at very low concentrations, often below the LoD, which complicated the analysis. Secondly, the IVOC factor is sometimes influenced by ethane, and due to the long atmospheric lifetime of ethane compared to that of NO and NO₂, the correlation with NOx depends on the age of the ethane measured. As indicated in lines 483–485, a link exists between the IVOC factor and combustion processes; however, it is challenging to distinguish a specific combustion process, as chemical processes also impact the partitioning of IVOCs between the particulate and gaseous phases. These factors collectively influence the correlation with NOx.*

*Other subsets of VOCs, such as those associated with traffic exhaust (Table 5), fuel evaporation (Table 6), and residential heating (Table 7), showed a better correlation with NO₂ (and NO during wintertime).*

*Although the daytime analysis was not shown, based on Figure 13 (now figure 5 in the new draft) (see below the figure but with only the IVOC factor), the IVOC factor appears to be influenced by the planetary boundary layer (PBL) during summertime, with maxima occurring during nighttime and minima during daytime. This influence was not observed in winter, where two distinct peaks were observed. The diurnal profile remains similar when considering the sum of IVOCs (see the second figure below). These observations suggest that, despite ethane driving the factor in certain seasons, the diurnal cycle of this factor is representative of the diurnal cycle of IVOCs. Furthermore, the wintertime profile and the correlation with both NOx and other PMF factors associated with combustion processes indicate that the IVOC factor is linked to combustion-related processes. These observations are consistent with the results obtained from the correlation table, which is why they were not explicitly shown in the article.*

*Finally, regarding the omission of ozone, the objective of the correlational analysis was to identify whether any link existed between the IVOC factor and various combustion processes. Since the ozone profile is more closely linked to chemical processes rather than direct combustion, we concluded that ozone was not pertinent to this analysis.*

[Figure]

**Figure 1: Figure 5 in the new draft of the article modified to show only the diurnal profile of the IVOC factor**

[Figure]

**Figure 2: Diurnal profile of the sum of IVOC**

5. **Biogenic Sources**: Figure 13 is excellent, showing traffic peaks and seasonal contributions from different sources. However, I was surprised that the biogenic factor appeared only in summer. Given the vegetative nature of the region, I would expect biogenic contributions during spring and fall as well. Could it be that the biogenic contributions were merged with another factor? It would be interesting to know which factor monoterpenes and isoprene were assigned to.

*Firstly, concerning the isoprene, we had some issues during fall 2019 leading to invalidate the data. To be more precise, there was a strong artefact near the isoprene that was anti-*

*correlated with the temperature. Unfortunately, during fall 2019 the isoprene was often co-eluted with this artefact. Also, as indicated lines 213 – 214 the alkenes were invalidated for spring 2020. The isoprene was explained at 80% by the local and regional urban background factor in spring 2019 and the monoterpenes at 60 – 70% for both spring 2019 and spring 2020. For this season the hypothesis is that the biogenic factor was present but not strong enough to be separated from the background. For this concern the following sentences were added line 435* "It is interesting to note that during springs the monoterpenes (and isoprene for spring 2019) are mainly explained by this factor. A possible explanation is that biogenic emissions were detected during these seasons but not strongly enough to be well identified by the PMF analysis."

*Concerning winter 2020 the isoprene is explained at 90% by fuel evaporation from traffic which is an interesting result as traffic can be the major contributor to isoprene emissions in cities in winter (Borbon et al., 2001; Panopoulou et al., 2020) and the following sentences are added line 318* "It is interesting to note that in winter 90% of isoprene is explained by this factor. Some studies have already observed the traffic as the major contributor to isoprene emissions in cities in winter (Borbon et al., 2001; Panopoulou et al., 2020)". *As for the monoterpenes their concentrations are mainly explained by the residential heating (30 – 60%) with also a part explained by the IVOC factor (10 – 40%) as indicated lines 364 – 365.*

*Finally, concerning fall 2019 the terpenes were mainly explained by the IVOC factor (70%) followed by the fuel evaporation factor (30%). This observation means that the monoterpenes measured here are probably from combustion sources but without the possibility to distinguish some specifics sources.*

*To resume globally and based on the results we have, the biogenic source was also observed during springs but was not strong enough to be distinguished from the local and regional urban background. As for fall and winter isoprene and monoterpenes were mainly explained by combustion sources such as traffic or residential heating.*

6. **Unconventional VOC Sources**: The manuscript provides a valuable regional discussion of VOC impacts on SOAPs and OFPs, highlighting traffic as a major contributor. However, other less conventional sources - such as cooking (restaurants, fast food), asphalt, and human emissions (e.g., consumer care products) - weren't discussed. These could be significant contributors, and I think it would strengthen the manuscript to briefly mention or speculate on these potential sources.

*Indeed, we based the SOAPs and OFPs results on the sources we were able to identify but some sources were not mentioned despite the high impact they can have on air quality because we did not have any tracer of these sources or they were hidden due to their low contribution to VOC emission in comparison to the sources identified.*

*Concerning asphalt related source, a recent study by Lasne et al. (2023) revealed that asphalt emissions in Paris were estimated at 2.9% of the total VOC emissions, and their contribution to total $PM_1$ emissions from inventoried sources was around 0.32% to 0.82%. The study also noted that these results represent a lower limit, and that actual asphalt emissions under real-world conditions are higher (Lasne et al., 2023). Additionally, a study by Lostier et al., 2025 indicates that IVOCs constitute a significant fraction of asphalt emissions. Therefore, although this source was not clearly identified in our study, it could potentially explain part of the IVOC concentrations measured, especially during summer when temperatures are higher.*

*Regarding cooking, few studies focus specifically on Europe. The emitted VOCs from cooking are primarily alcohols and heavy aldehydes (Kumar et al., 2025), which were not measured in our study. This source was estimated to contribute to 0.69% of the yearly VOC emissions in the UK (Carter et al., 2024).*

*Finally, concerning Volatile Chemical Products (VCPs), which include personal care products, cleaning agents, pesticides, etc., it has been estimated that around half of VOC emissions in both the U.S. and Europe are attributable to VCPs (McDonald et al., 2018). However, this source was not identified in our study for several reasons:*

1. *VCPs encompass a broad range of potential sources with distinct chemical fingerprints, each exhibiting different diurnal and seasonal variations.*
2. *While some aromatics, such as toluene or xylenes, may be associated with VCP use, they are primarily linked to petroleum distillates, which constitute only a portion of VCPs. Many compounds, such as ethanol, acetone, siloxanes, and glycols, also present in VCP emissions, were not measured in our study. Therefore, measuring only NMHCs (non-methane hydrocarbons) leads to an underestimation or non-identification of VCPs as a source.*

*By combining consumer and industrial VCPs, they are responsible for 61% of the SOAFP, according to McDonald et al. (2018).*

*Therefore, the following sentences are added at line 550: "The results presented above are limited to the sources that have been detected with the measurement of NMHCs. Many sources not identified here can significantly impact the formation of secondary pollutants. Asphalt, which covers a substantial portion of urban ground surfaces, contributes about 3% of total VOC emissions in Paris according to Lasne et al. (2023). Since the main VOCs detected were heavy alkanes and aromatics (Lostier et al., 2025), we can infer that this source likely contributes significantly to the SOAFP.*

*Another source not discussed here is cooking. This source, composed mainly of alcohols and heavy aldehydes (Kumar et al., 2024), contributes a minor fraction of VOC emissions (0.69% of yearly VOC emissions in the UK according to Carter et al., 2024). However, since aldehydes are strong ozone precursors according to MIR values (Venececk et al., 2018), cooking may have a potential impact on the OFP in urban environments.*

*Lastly, regarding solvent use, which was not identified in this study despite its substantial contribution to VOC emissions inventories, it is common to either underestimate its contribution or fail to identify it when measuring only NMHCs (McDonald et al., 2018). Incorporating oxygenated VOCs, such as alcohols, acetone, glyoxals, or siloxanes, into the measurement dataset would greatly aid in identifying this source. It has been estimated that in both the U.S. and Europe, this source contributes to 61% of the SOAFP (McDonald et al., 2018), underscoring the need for better characterization of this source in urban ambient air environments."*

7. **Biogenic VOCs (BVOCs)**: While the source apportionment work seems solid overall, the discussion of biogenic VOCs (BVOCs) seems somewhat underplayed. Given that BVOCs are important in Mediterranean coastal regions, it might be useful to include a

brief mention of sea breeze effects or biogenics from seawater, as reported in other studies (e.g., Dayan et al., 2020).

*We focused on the biogenic emission from the park as it is our main source based on our pollution rose but when in sea breeze condition we cannot totally discard BVOCs from seawater. Then the following sentences were added line 437. "However, in sea breeze condition we cannot discard marine origins for this biogenic source. Especially when the main driver of this source is the isoprene. This compound has already been observed with high concentrations, until 9 ppb, coming from the sea in another Mediterranean coastal city (Dayan et al., 2020). Then, considering the lifetime of isoprene the park close to the station remains the major emission area for BVOC but biogenic emissions from the sea cannot be discarded."*

Overall, this paper is informative and provides valuable insights into VOC sources in an urban Mediterranean setting. I recommend reorganizing the manuscript to create a more cohesive story with fewer figures in the main text (moving the rest to SI). This would help make the paper more accessible to a wider audience.

Borbon, A., Fontaine, H., Veillerot, M., Locoge, N., Galloo, J. C., and Guillermo, R.: An investigation into the traffic-related fraction of isoprene at an urban location, Atmos. Environ., 35, 3749–3760, https://doi.org/10.1016/S1352-2310(01)00170-4, 2001.

Carter, T. J., Shaw, D. R., Carslaw, D. C., and Carslaw, N.: Indoor cooking and cleaning as a source of outdoor air pollution in urban environments, Environ. Sci. Process. Impacts, 26, 975–990, https://doi.org/10.1039/D3EM00512G, 2024.

Cui, L., Wu, D., Wang, S., Xu, Q., Hu, R., and Hao, J.: Measurement report: Ambient volatile organic compound (VOC) pollution in urban Beijing: characteristics, sources, and implications for pollution control, Atmospheric Chem. Phys., 22, 11931–11944, https://doi.org/10.5194/acp-22-11931-2022, 2022.

Dayan, C., Fredj, E., Misztal, P. K., Gabay, M., Guenther, A. B., and Tas, E.: Emission of biogenic volatile organic compounds from warm and oligotrophic seawater in the Eastern Mediterranean, Atmospheric Chem. Phys., 20, 12741–12759, https://doi.org/10.5194/acp-20-12741-2020, 2020.

Debevec, C., Sauvage, S., Gros, V., Salameh, T., Sciare, J., Dulac, F., and Locoge, N.: Seasonal variation and origins of volatile organic compounds observed during 2 years at a western Mediterranean remote background site (Ersa, Cape Corsica), Atmospheric Chem. Phys., 21, 1449–1484, https://doi.org/10.5194/acp-21-1449-2021, 2021.

Derwent, R. G., Jenkin, M. E., Pilling, M. J., Carter, W. P. L., and Kaduwela, A.: Reactivity Scales as Comparative Tools for Chemical Mechanisms, J. Air Waste Manag. Assoc., 60, 914–924, https://doi.org/10.3155/1047-3289.60.8.914, 2010.

Frischmon, C. and Hannigan, M.: VOC source apportionment: How monitoring characteristics influence positive matrix factorization (PMF) solutions, Atmospheric Environ. X, 21, 100230, https://doi.org/10.1016/j.aeaoa.2023.100230, 2024.

Gkatzelis, G. I., Coggon, M. M., McDonald, B. C., Peischl, J., Gilman, J. B., Aikin, K. C., Robinson, M. A., Canonaco, F., Prevot, A. S. H., Trainer, M., and Warneke, C.: Observations Confirm that Volatile Chemical Products Are a Major Source of Petrochemical Emissions in U.S. Cities, Environ. Sci. Technol., 55, 4332–4343, https://doi.org/10.1021/acs.est.0c05471, 2021.

Holland, R., Khan, A. H., Derwent, R. G., Lynch, J., Ahmed, F., Grace, S., Bacak, A., and Shallcross, D. E.: Gas-phase kinetics, POCPs, and an investigation of the contributions of VOCs to urban ozone production in the UK, Int. J. Chem. Kinet., 55, 350–364, https://doi.org/10.1002/kin.21640, 2023.

Hopke, P. K., Chen, Y., Rich, D. Q., Mooibroek, D., and Sofowote, U. M.: The application of positive matrix factorization with diagnostics to BIG DATA, Chemom. Intell. Lab. Syst., 240, 104885, https://doi.org/10.1016/j.chemolab.2023.104885, 2023.

In 'T Veld, M., Seco, R., Reche, C., Pérez, N., Alastuey, A., Portillo-Estrada, M., Janssens, I. A., Peñuelas, J., Fernandez-Martinez, M., Marchand, N., Temime-Roussel, B., Querol, X., and Yáñez-Serrano, A. M.: Identification of volatile organic compounds and their sources driving ozone and secondary organic aerosol formation in NE Spain, Sci. Total Environ., 906, 167159, https://doi.org/10.1016/j.scitotenv.2023.167159, 2024.

Kumar, A., O'Leary, C., Winkless, R., Thompson, M., Davies, H. L., Shaw, M., Andrews, S. J., Carslaw, N., and Dillon, T. J.: Fingerprinting the emissions of volatile organic compounds emitted from the cooking of oils, herbs, and spices, Environ. Sci. Process. Impacts, 27, 244–261, https://doi.org/10.1039/D4EM00579A, 2025.

Lasne, J., Lostier, A., Romanias, M. N., Vassaux, S., Lesueur, D., Gaudion, V., Jamar, M., Derwent, R. G., Dusanter, S., and Salameh, T.: VOC emissions by fresh and old asphalt pavements at service temperatures: impacts on urban air quality, Environ. Sci. Atmospheres, 3, 1601–1619, https://doi.org/10.1039/D3EA00034F, 2023.

Lostier, A., Sarica, T., Lasne, J., Roose, A., Sartelet, K., Jamar, M., Gaudion, V., Dusanter, S., Lesueur, D., Chen, H., Salameh, T., and Romanias, M. N.: Real-World Asphalt Pavement Emissions: Combining Simulation Chamber Measurements and City Scale Modeling to Elucidate the Impacts on Air Quality, ACS EST Air, acsestair.4c00323, https://doi.org/10.1021/acsestair.4c00323, 2025.

Mai, J.-L., Yang, W.-W., Zeng, Y., Guan, Y.-F., and Chen, S.-J.: Volatile organic compounds (VOCs) in residential indoor air during interior finish period: Sources, variations, and health risks, Hyg. Environ. Health Adv., 9, 100087, https://doi.org/10.1016/j.heha.2023.100087, 2024.

Putaud, J.-P., Pozzoli, L., Pisoni, E., Martins Dos Santos, S., Lagler, F., Lanzani, G., Dal Santo, U., and Colette, A.: Impacts of the COVID-19 lockdown on air pollution at regional and urban background sites in northern Italy, Atmospheric Chem. Phys., 21, 7597–7609, https://doi.org/10.5194/acp-21-7597-2021, 2021.

Simon, L.: Détermination des sources de composés organiques (gazeux et particulaires) en Ile-de-France, 2023.